# Breakdown of the velocity and turbulence in the wake of a wind turbine - Part 1: large eddy simulations study.

Erwan Jézéquel[1,2], Frédéric Blondel[1], and Valéry Masson[2]

[1]IFP Energies nouvelles, 1-4 Avenue de Bois Préau, Rueil-Malmaison, France
[2]Centre National de Recherches Météorologiques, 42 avenue Gaspard Coriolis, Toulouse, France

**Correspondence:** Erwan Jézéquel (erwan.jezequel@ifpen.fr)

**Abstract.** A new theoretical framework, based on an analysis in the moving and fixed frames of reference (MFOR and FFOR), is proposed to break down the velocity and turbulence fields in the wake of a wind turbine. This approach adds theoretical support to models based on the dynamic wake meandering (DWM) and opens the way for a fully analytical and physically-based model of the wake that takes meandering and atmospheric stability into account, which is developed in the companion paper. The mean velocity and turbulence in the FFOR are broken down into different terms, which are functions of the velocity and turbulence in the MFOR. These terms can be regrouped as pure-terms and cross-terms. In the DWM, the former group is modelled and the latter is implicitly neglected. The shape and relative importance of the different terms are estimated with the large eddy simulation solver Meso-NH coupled with an actuator line method. A single wind turbine wake is simulated on flat terrain, under three cases of stability: neutral, unstable, and stable. In the velocity breakdown, the cross-term is found to be relatively low. It is not the case for the turbulence breakdown equation where even though the cross-terms are overall of lesser magnitude than the pure-terms, they redistribute the turbulence and induce a non-negligible asymmetry. These findings underline the limitations of models that assume a steady velocity in the MFOR, such as the DWM or the model developed in the companion paper. It is also found that as atmospheric stability increases, the pure turbulence contribution becomes relatively larger and pure meandering relatively smaller.

## 1 Introduction

The wake behind a wind turbine is characterised by a decrease of wind velocity and an increased level of turbulence compared to the inflow properties, leading respectively to a decreased generated power and increased loads for downstream turbines. The stability of the atmospheric boundary layer (ABL) influences the wake recovery (Abkar and Porté-Agel, 2015) and the large-scale eddies carried in this region of the atmosphere induce wake meandering, i.e. oscillations of the instantaneous wake around its mean position (Larsen et al., 2008). This phenomenon is schematised in Fig. 1: the instantaneous wake at two different times is drawn in blue, and the time-averaged wake is drawn in red. The meandering can cause a downstream turbine to be successively outside inside (a) and outside (b) the wake even though on a time-averaged basis it is always fully embedded in the wake (in red in both schemes). Due to these unsteady displacements, the time-averaged wake widths will be larger and

the time-averaged maximum velocity deficit lower (dashed red curve in Fig. 1) than if there was no meandering (dashed blue
curve in Fig. 1).

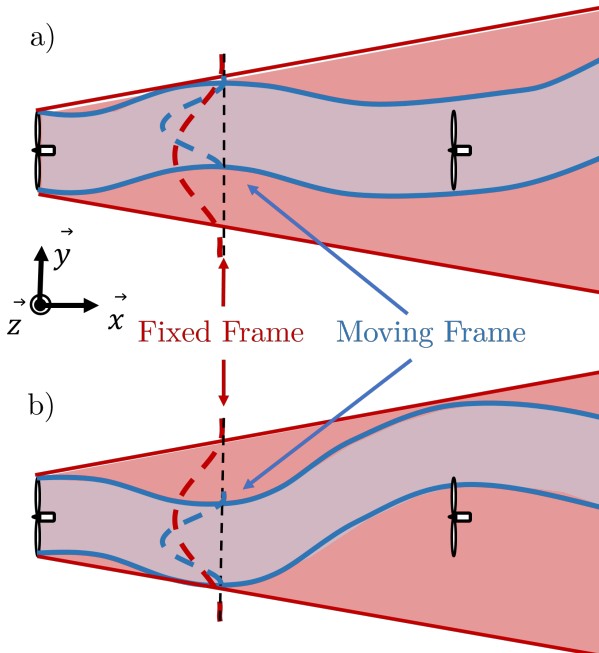

**Figure 1.** Schematic of the wake meandering phenomenon. The mean (red) and instantaneous (blue) wake outlines are plotted at two different time steps: a) the downstream turbine is inside the wake; b) the downstream turbine is partially outside the wake. The mean velocity profiles in the fixed frame (red) and the moving frame (blue) are also plotted in dashed lines.

   The evolution of the time-averaged wake may thus be considered as the combination of two phenomena: on one hand the wake expansion and dissipation due to the turbulent diffusion and on the other hand the wake meandering due to large-scale forcing of the ABL. Most analytical models are calibrated directly in the frame of reference linked to the ground (called hereafter fixed frame of reference or FFOR) against reference data averaged over the meandering time-period, and the wake
widths are written as a function of the turbulence intensity (TI) upstream the turbine (Fuertes et al., 2018; Ishihara and Qian, 2018). This approach is straightforward but the phenomena of meandering and turbulent mixing are not differentiated. The issue is that the atmospheric stability impacts meandering, leading to different time-averaged wake recoveries for a given upstream TI at hub height (Du et al., 2021). In order to model accurately wind turbine wakes in non-neutral ABL, it is proposed to decouple the effect of meandering and the effect of wake expansion.
This can be achieved with the use of the moving frame of reference (MFOR), which is moving with the wake centre at each time step. The unsteady velocity field in the MFOR is thus equivalent to the velocity field that would be observed if there was no meandering. Due to the spreading caused by the meandering, the mean velocity deficit in the FFOR is weaker and wider compared to the mean velocity deficit in the MFOR (red and blue dashed profiles in Fig 1). Conversely, the turbulence (not

shown on the scheme) is stronger in the FFOR compared to the MFOR (Larsen et al., 2019). If a Cartesian coordinate system $(x, y, z)$ is used for the streamwise, lateral and vertical coordinates respectively (see Fig. 1), the instantaneous velocity can be changed from one frame to another according to the relation:

$$U_{MF}(x, y, z, t) = U_{FF}(x, y + y_c(x, t), z + z_c(x, t), t) \tag{1}$$

where subscripts MF and FF denote the velocity fields in the MFOR and FFOR respectively and $y_c(x, t)$ and $z_c(x, t)$ are the time series of the wake centre's coordinates at the downstream position $x$. The concept of MFOR has originally been introduced for the dynamic wake meandering (DWM) model (Larsen et al., 2008) which aims at modelling the unsteady effects of meandering. The methodology to retrieve the velocity and turbulence fields in the FFOR with this model is briefly introduced here. In the DWM model, the wake in the MFOR is assumed to be steady and axisymmetric (Ainslie, 1985), and wake expansion and dissipation are assumed to be driven by turbulent mixing and the turbine's operating conditions. This steady wake is advected as a passive tracer by the largest eddies of the ABL to get the unsteady wake in the FFOR. If the unsteady FFOR velocity field is required, Eq. 1 is used with a steady, axisymmetric form in the MFOR, i.e. $U_{FF}(x, y, z, t) = U_{MF}(x, y - y_c(x, t), z - z_c(x, t))$. If only the time-averaged field is needed, Eq. 1 reduces to a 2D convolution product (Keck et al., 2013b), denoted $**$ in the following. This is possible in the DWM framework since $U_{MF,dwm}$ is considered to be steady and thus the elements of the wake centre time series can be permuted without affecting the results of Eq. 1. It gives:

$$\overline{U_{FF,dwm}}(y, z) = U_{MF,dwm}(y, z) ** f_c(y, z) = \int \int U_{MF,dwm}(y - y_c, z - z_c) f_c(y_c, z_c) dy_c dz_c \tag{2}$$

where $f_c(y, z)$ is the probability density function (PDF) of the wake centre position, normalised such as $\int \int f_c(y_c, z_c) dy_c dz_c = 1$. Here and in the following, the Reynolds decomposition is used to write any unsteady field $X(t)$ as a sum of a mean and a fluctuating part: $X(t) = \overline{X} + X'(t)$.

In the DWM, the total turbulence (defined as the temporal variance of the velocity field) in the FFOR in the wake can be computed as the sum of two components:

$$k_{FF,dwm}(x, y, z) = k_{a,dwm}(x, y, z) + k_{m,dwm}(x, y, z) \tag{3}$$

where $k_a$ is the rotor-added turbulence, mainly driven by the shear generated by the velocity deficit in the wake and $k_m$ is the meandering turbulence, generated by the lateral and vertical displacements of the wake. Similarly to Eq. 2, these two components can be written (Keck et al., 2013b):

$$k_{a,dwm}(y,z) = \int\int k_{MF,dwm}(y-y_c, z-z_c) f_c(y_c, z_c) dy_c dz_c = k_{MF,dwm}(y,z) ** f_c(y,z) \tag{4}$$

$$k_{m,dwm}(y,z) = \int\int \left( U_{MF,dwm}(y-y_c, z-z_c) - \overline{U_{FF,dwm}}(y,z) \right)^2 f_c(y_c, z_c) dy_c dz_c \tag{5}$$

$$= U_{MF,dwm}^2(y,z) ** f_c(y,z) - \overline{U_{FF,dwm}}^2(y,z) \tag{6}$$

where $k_{MF,dwm}$ is the modelled turbulence in the MFOR, i.e. the turbulence that would be measured if there was no meandering. In the DWM model, an empirical scaling of $U_{MF}(y,z)$ with a factor $k_{mt}(y,z)$ is used to compute $k_{MF,dwm}$ (Madsen et al., 2010; Conti et al., 2021). Equation 6 is obtained by developing Eq. 5 and simplifying with Eq. 2. The added value of such an approach is that it allows writing the velocity and the turbulence in the FFOR as a function of the same fields in the MFOR, where they are presumed to be only dependent on the turbine's operating conditions, thus less complex and easier to model.

The objective of this work is to write the velocity and turbulence in the FFOR as a function of the velocity and turbulence in the MFOR and show the underlying DWM assumptions that neglect some terms. The importance of these missing terms for both velocity and turbulence is evaluated. The reference results come from large eddy simulations (LESs) of an isolated wind turbine wake over flat terrain. Three cases of stability, approximately corresponding to the SWiFT benchmark (Doubrawa et al., 2020), are simulated using Meso-NH (Lac et al., 2018) with an actuator line method (ALM) (Joulin et al., 2020; Jézéquel et al., 2021).

This work is separated into two articles. In the present one, the breakdown of the velocity and turbulence is presented and applied to the LESs datasets. In the companion paper, the results are used to build a new analytical model for velocity and turbulence in the wake of a wind turbine. The first part of the present article is dedicated to the development of the velocity and turbulence breakdowns, i.e. the expression of the velocity and turbulence fields in the FFOR as a function of their counterparts in the MFOR. In the second part, the numerical framework is detailed: it describes the SWiFT cases, the LES code Meso-NH, the numerical setup, the wake tracking algorithm and the limitations of these tools. In the third part, the LES datasets are used to quantify the error induced by the approximations necessary to write Eqs. 2 and 3. In the fourth part, some physical interpretations are proposed, the dependence of $k_a$ and $k_m$ on atmospheric stability is studied and the shape of all the terms in the turbulence breakdown equation is described.

## 2 Analytical development

To lighten the mathematical formulations, the notation $\widehat{a(y,z)} = a(y - y_c(t), z - z_c(t))$ will be used to express the switch between FFOR and MFOR (Eq. 1). This operation can be interpreted as an unsteady translation of any field $a$ by the meandering: the stronger the meandering, the more spread will be $\widehat{a}$. It is important to note that $a$ can be steady or unsteady, but $\widehat{a}$ is always unsteady. For any variables $a$ and $b$, the following properties hold:

$$\widehat{a} + \widehat{b} = \widehat{a+b}. \tag{7}$$

$$\widehat{a} \cdot \widehat{b} = \widehat{a \cdot b}. \tag{8}$$

$$\widehat{\overline{a}} \neq \overline{\widehat{a}}. \tag{9}$$

$$\overline{\overline{a}} = \overline{a} * * f_c. \tag{10}$$

Properties 7 and 8 are obtained from the linearity of the translation operator. Property 9 is trivial since $\widehat{\overline{a}}$ is time-dependent and $\overline{\widehat{a}}$ is not. Property 10 can be demonstrated by defining $f_c$ as a sum of indicator functions and applying a Taylor development. Using the $\widehat{\phantom{a}}$ notation and applying the Reynolds decomposition to $U_{MF}$ allows one to re-write Eq. 1 as:

$$U_{FF} = \widehat{U_{MF}} = \widehat{\overline{U_{MF}}} + \widehat{U'_{MF}} \tag{11}$$

The mean velocity in the FFOR can directly be deduced by applying the averaging operator to this equation:

$$\overline{U_{FF}} = \underbrace{\overline{\widehat{\overline{U_{MF}}}}}_{(I)} + \underbrace{\overline{\widehat{U'_{MF}}}}_{(II)} \tag{12}$$

From Eq. 10, it appears that the term (I) is the convolution of $\overline{U_{MF}}$ with $f_c$, and can be viewed as a pure mean velocity term: it is null only if the mean velocity is null. Conversely, the term (II) is here viewed as a cross-term because it can be equal to $0$ either if there is no meandering ($\widehat{x} = x$) or if there is no turbulence in the MFOR ($U'_{MF} = 0$). In the DWM model, $U_{MF,dwm}$ is steady so $U_{MF,dwm} = \overline{U_{MF,dwm}}$ and $U'_{MF,dwm} = 0$, thus Eqs. 2 and 12 are equivalent. The assumption of steady flow in the MFOR for analytical or DWM models is equivalent to the assumption that term (II) of Eq. 12 is negligible. Since $U'_{MF} = 0$ is not necessarily true in real cases (nor in LESs, which are used here) this hypothesis must be verified, which is one of the objectives of the present work.

For the turbulence equation, one can write from Eqs. 11 and 12:

$$\overline{U_{FF}^2} = \overline{\widehat{\overline{U_{MF}}}^2} + 2\overline{\widehat{\overline{U_{MF}}}\widehat{U'_{MF}}} + \overline{\widehat{U'_{MF}}^2} \tag{13}$$

$$\overline{U_{FF}}^2 = \overline{\widehat{\overline{U_{MF}}}}^2 + 2\overline{\widehat{\overline{U_{MF}}}}\,\overline{\widehat{U'_{MF}}} + \overline{\widehat{U'_{MF}}}^2 \tag{14}$$

The total turbulence in the FFOR can then be written as a function of the preceding quantities:

$$k_{FF} = \overline{U_{FF}'^2} = \overline{U_{FF}^2} - \overline{U_{FF}}^2$$

$$= \overline{\widehat{\widehat{U_{MF}}}^2} - \overline{\widehat{\widehat{U_{MF}}}}^2 + 2\left(\overline{\widehat{\widehat{U_{MF}}}\widehat{U_{MF}'}} - \overline{\widehat{\widehat{U_{MF}}}}\,\overline{\widehat{U_{MF}'}}\right) + \overline{\widehat{U_{MF}'}^2} - \overline{\widehat{U_{MF}'}}^2$$

$$= \underbrace{\overline{\widehat{\widehat{U_{MF}}}^2} - \overline{\widehat{\widehat{U_{MF}}}}^2}_{(III)} + \underbrace{2\mathrm{cov}\left(\overline{\widehat{\widehat{U_{MF}}}},\overline{\widehat{U_{MF}'}}\right)}_{(V)} + \underbrace{\overline{\widehat{\widehat{U_{MF}'^2}}}}_{(IV)} + \underbrace{\overline{\left(\widehat{U_{MF}'^2}\right)'}}_{(VI)} - \underbrace{\overline{\widehat{U_{MF}'}}^2}_{(VII)}$$

$$= \underbrace{k_m}_{(III)} + \underbrace{k_a}_{(IV)} + \underbrace{2\mathrm{cov}\left(\overline{\widehat{\widehat{U_{MF}}}},\overline{\widehat{U_{MF}'}}\right)}_{(V)} + \underbrace{\overline{\left(\widehat{U_{MF}'^2}\right)'}}_{(VI)} - \underbrace{\overline{\widehat{U_{MF}'}}^2}_{(VII)} \tag{15}$$

The term (III), also written $k_m$ in the following for consistency with the literature (Keck et al., 2013b; Conti et al., 2021), is the turbulence purely induced by meandering: in the case of a meandering steady wake, i.e. $U_{MF}' = 0$, Eq. 15 reduces to
this term only. Note that $k_m$ is the variance of $\widehat{\widehat{U_{MF}}}$. In the DWM model, the wake in the MFOR is steady, but a rotor-added turbulence term is added to model the small-scale turbulence that exists in the MFOR in real cases. This rotor-added turbulence can be calibrated from the MFOR turbulence in reference data i.e. term (IV) of Eq. 15. It is the turbulence purely induced by the rotor: in absence of meandering ($\widehat{x} = x$), the equation reduces to this term only, also written $k_a$ in the following. It is also denoted $k_a$ for consistence with the literature.

Through Eq. 3, it is assumed in the DWM that the wake turbulence is separated between two terms: one purely induced by the meandering ($k_{m,dwm}$, related to term (III)) and the other purely induced by the rotor ($k_{a,dwm}$, related to term (IV)). The analysis presented above shows that three cross-terms are neglected under this assumption. Term (V) is the covariance of $\widehat{\widehat{U_{MF}}}$ and $\widehat{U_{MF}'}$, term (VI) is the remaining of $\overline{\widehat{U_{MF}'}^2}$ when subtracting the rotor-added turbulence in the FFOR $k_a = \overline{\widehat{\widehat{U_{MF}'^2}}}$ (it can be viewed as the varying part of the MFOR turbulence) and term (VII) is the square of the term (II). It is a dissipation term as it is
always negative. Like the term (II), these are cross-terms since they are equal to zero if either the turbulence in the MFOR or the meandering is null.

Similarly to the velocity field, Eq. 15 shows that when calibrating a DWM-type model against realistic data (measurement or high-fidelity simulation, denoted $._{cal}$), if it is assumed that $U_{MF,dwm} = \overline{U_{MF,cal}}$ and $k_{MF,dwm} = \overline{U_{MF,cal}'^2}$, then there will be three missing terms: (V), (VI) and (VII). Like term (II) for the velocity, these terms cannot be computed directly from a
steady model of the velocity in the DWM so similarly to term (IV), they must be modelled differently.

It has thus been shown in this section that the mean velocity turbulence fields in the FFOR can be broken down into two types of terms: pure-terms ((I), (III) and (IV)) and cross-terms ((II), (V), (VI) and (VII)). In models where the wake is considered steady in the MFOR and advected as a passive tracer (such as the DWM or the model developed in the companion paper), the pure-terms are modelled but the cross-terms are implicitly neglected. The error induced by this assumption is verified in this
work with LESs.

## 3 Methodology

### 3.1 The SWiFT benchmark

The breakdowns of the mean velocity and turbulence fields in the FFOR described in Sect. 2 are applied to three LESs cases. These datasets, already presented in Jézéquel et al. (2022), are the result of simulations that reproduce the SWiFT benchmark (Doubrawa et al., 2020) with the LES code Meso-NH (Lac et al., 2018). The simulated turbine is a modified version of the Vestas V27: it is a three-bladed rotor with a diameter of $D = 27$ m and a hub height of 32.1 m. The orography of the terrain is neglected, and three cases of stability are simulated: near-neutral, unstable and strongly stable. The simulations are classified with the Monin-Obukhov length:

$$L_{MO} = -\frac{u_*^3 \theta}{\kappa g \overline{\theta' w'}} \tag{16}$$

where $u_* = (\overline{u'w'}^2 + \overline{v'w'}^2)^{1/4}$ is the friction velocity, $\overline{\theta' w'}$ is the turbulent potential temperature flux, and $\theta$ is the potential temperature. All these variables are computed at $z = 10$ m above the ground. $\kappa$ and $g$ are the Von Kármán and earth gravity constants. For the neutral, unstable and stable cases, the stability parameters at $z = 10$ m are respectively $z/L_{MO} = \{0.003, -0.16, 0.60\}$, the inflow velocities at hub height are $U_h = \{8.4, 6.2, 4.7\}$ m s$^{-1}$, the inflow streamwise turbulence intensities at hub height are $TI_x = \sqrt{k_x}/\overline{U}_x = \{11.2, 12.3, 3.7\}$ % ($k_x$ is the variance of $U_x$) and the thrust coefficients are $C_T = \{0.79, 0.81, 0.82\}$.

### 3.2 The Meso-NH LES solver

Meso-NH (MESOscale Non Hydrostatic) is a finite volume and finite difference, open-source research code for ABL simulations developed by the Centre National de Recherches Météorologiques and the Laboratoire d'Aérologie. The model is described in detail in Lafore et al. (1998) and Lac et al. (2018). The filtered Navier-Stokes and energy conservation equations are resolved on an Arakawa C-grid. The unknowns of the system are the velocities ($U_x$, $U_y$ and $U_z$) and the potential temperature $\theta$. A constant density profile $\rho(z)$ is imposed, except for the buoyancy term (anelastic assumption) and the vertical velocity is driven by the vertical pressure gradient and the gravity (non-hydrostatic set of equations). The Coriolis force is added to the momentum equation, as well as a large-scale forcing term, which is imposed by the user through a 2-D geostrophic wind $U_g$.

The turbulence closure is of order 1.5: an additional equation is introduced for the subgrid kinetic energy $e_{sgs}$ and the other subgrid terms are modelled as functions of the resolved quantities, $e_{sgs}$ and a mixing length $L_m$ (Cuxart et al., 2000). The mixing length is related to the grid size and stratification through the Deardorff formulation (Deardorff, 1980). This set of equations is discretised spatially with a fourth-order centred scheme and temporally with a fourth-order Runge-Kutta scheme.

To model the wind turbine, the ALM is used, following Sørensen and Shen (2002). This method has been implemented in Meso-NH, validated against the NewMexico wind tunnel experiments (Joulin et al., 2020) and the *in situ* measurements and LESs codes of the SWiFT benchmark (Jézéquel et al., 2021). A grid nesting technique allows the coupling of two or more computational domains of different sizes, temporal and spatial resolutions (Stein et al., 2000). The velocity field of a father

domain $D_i$ is interpolated to the boundaries of a son domain $D_{i+1}$. Hence, the resolution can be brought below the metre (necessary here to have 30 mesh points per blade as recommended in Troldborg (2009)), while still taking into account the large-scale behaviour of the ABL.

### 3.3 Numerical parameters

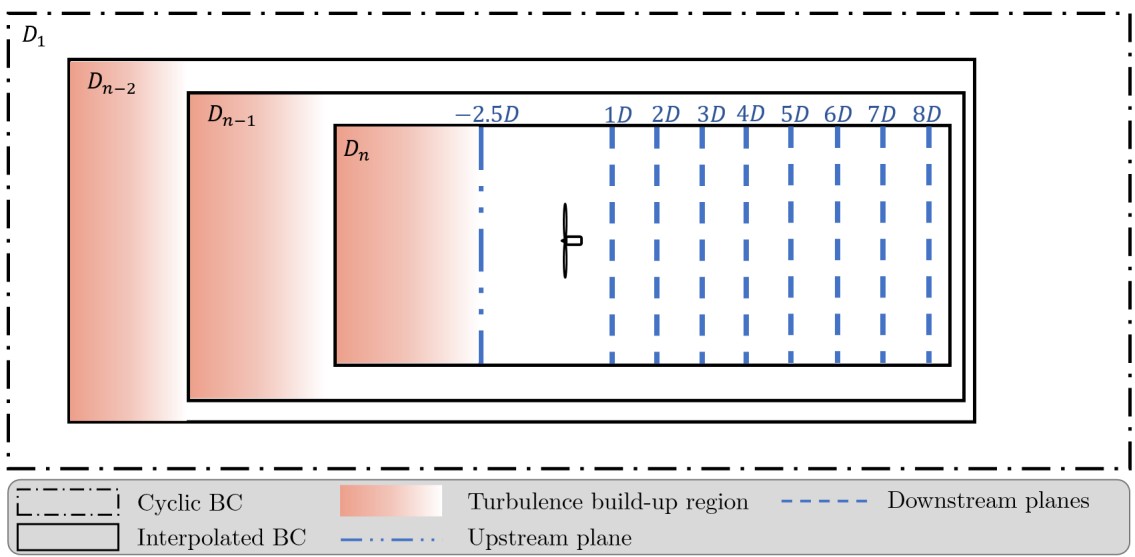

**Figure 2.** Schematic of the simulation set-up with Meso-NH.

The numerical parameters used for the three simulations are presented in Table 1 for the different domains of the grid nesting. The size of the horizontal mesh depends on the domain $D_i$ but in Meso-NH the vertical mesh is the same for every domain. In the induction and the wake regions, the vertical discretisation $\Delta Z$ is set to have isotropic cells in the most refined domain i.e. $D_4$ in the neutral and unstable cases and $D_2$ in the stable case. The bottom boundary condition is determined by the subgrid

heat $\overline{w'\theta'}$ and momentum $\overline{w'u'}$ fluxes. The heat flux is prescribed and governs the evolution of $\theta$ in the middle of the first cell, along with other resolved processes such as advection. The momentum flux at the surface is computed according to the Monin-Obukhov similarity laws, depending on the roughness length, wind at the middle of the first grid mesh and heat flux. It is used to compute the velocity at the first grid mesh.

The flowfield is initialised with a constant-velocity profile equal to the geostrophic wind. A constant-temperature profile

is set up to an arbitrary defined ABL height, capped by an inversion region of 5 K over a depth of 50 m. The geostrophic wind, ABL height, surface roughness $z_0$ and kinematic vertical heat flux are chosen to be as close as possible to the SWiFT measurements in terms of velocity, wind direction, TKE and stability parameter.

In the first domain $D_1$, the boundary conditions are cyclic to let the turbulence establish, with dimensions $L_X$ and $L_Y$ larger than the largest eddies of the flow. In a stable ABL, these eddies are smaller, which is why a smaller domain $D_1$ is suited, and

|  | Neutral | | | | Unstable | | | | Stable | |
|---|---|---|---|---|---|---|---|---|---|---|
|  | $D_1$ | $D_2$ | $D_3$ | $D_4$ | $D_1$ | $D_2$ | $D_3$ | $D_4$ | $D_1$ | $D_2$ |
| $z_0$ [mm] | 14 | | | | 14 | | | | 14 | |
| $\overline{w'\theta'}$ [K m s$^{-1}$] | –0.0020 | | | | 0.0247 | | | | -0.0047 | |
| ABL height [m] | 1000 | | | | 1000 | | | | 200 | |
| $U_g$ [m s$^{-1}$] | (u=11.42, v=-3.7) | | | | (u=8.1, v=-1.2) | | | | (u=7.6, v=-3.1) | |
| $\Delta Z$ [m] | 0.5 | | | | 0.5 | | | | 0.4 | |
| $\Delta X = \Delta Y$ [m] | 20 | 4 | 1 | 0.5 | 20 | 4 | 1 | 0.5 | 1.2 | 0.4 |
| $L_X$ [m] | 6000 | 3200 | 640 | 432 | 12000 | 4000 | 1080 | 500 | 540 | 480 |
| $L_Y$ [m] | 2400 | 1600 | 320 | 216 | 6000 | 2000 | 540 | 250 | 300 | 180 |
| $\Delta t$ [ms] | 200 | 100 | 50 | 8 | 100 | 100 | 50 | 10 | 140 | 11 |
| Simulation time [s] | 4800 | | | | 2400 | | | | 600 | |
| $\Omega$ [rad s$^{-1}$] | 4.56 | | | | 3.89 | | | | 2.79 | |
| $\gamma$ [deg] | –0.75 | | | | –0.75 | | | | –0.75 | |

**Table 1.** Numerical parameters used in Meso-NH.

inversely for the unstable case. After an initialisation of turbulence in domain $D_1$, the nested domains ($D_2$, $D_3$ and $D_4$) are successively created. In each nested domain $D_i$, a region in which the turbulent flow adapts to the finer resolution (in brown in Fig. 2) appears near the inflow. The next domain $D_{i+1}$ must avoid it, so a spectral analysis (not shown here) has been carried out to measure the end of this perturbed region. The time step in every domain is driven by the CFL condition, except for the finest domain, where it is equal to the time needed for the tip of the blades to cross one cell.

The size of the domain of interest ($D_4$ in the neutral and unstable case and $D_2$ in the stable case) is set to compute the wake up to 8 diameters downstream of the turbine. This choice has been made to keep reasonable simulation times for the LES and a high degree of confidence in the wake tracking algorithm. However, the wake is not dissipated at this position, and the present work could be completed with a study where the wake is computed further downstream, e.g. at $x/D = 15$.

The ALM is activated once the flow is established in the most refined domain, and after a 10 minutes spin-up to let the wake flow establish, the instantaneous velocity is extracted at one plane upwind of the turbine and several planes downwind, according to Fig. 2. Note that the simulation time is case-dependent: 80, 40 and 10 minutes for the neutral, unstable and stable cases respectively.

The rotational velocity of the wind turbine $\Omega$ and pitch of the blades $\gamma$ are set constant to a value interpolated in the controller table of the turbine with the upstream velocity at hub height $U_h$, and a simple implementation of the nacelle and the tower is used (corresponding to Stevens et al. (2018)).

### 3.4 Wake tracking

The wake meandering is characterised by the time series of the wake centre coordinates $y_c(x,t)$ and $z_c(x,t)$. Even though it is a very handy concept from a theoretical point of view, defining the centre of the wake or even its borders is difficult, especially when the wake is developing inside a turbulent boundary layer. Indeed, the turbulent structures can move, twist or even split the wake, and low-velocity eddies can be mistaken for the wake.

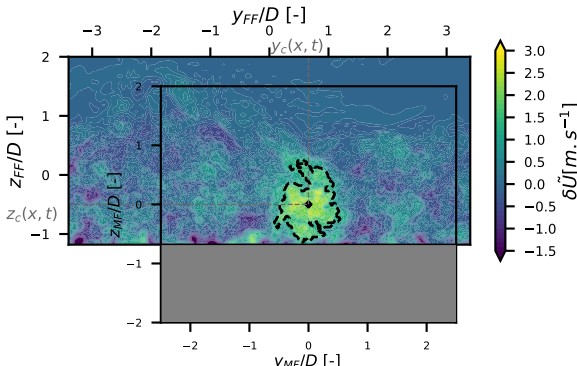

**Figure 3.** Result of the wake tracking at an arbitrary time step at x=6D downstream. The detected isocontour is in dashed line and the detected wake centre is represented by a diamond.

To determine the wake centre at each time step, an algorithm based on the conservation of momentum in the wake is used (Quon et al., 2020). First, the 2D velocity and momentum deficits are computed at each time step and for each downstream plane:

$$\delta\tilde{U} = \tilde{U}_{ref}(x,y,z,t) - \tilde{U}(x,y,z,t) \tag{17}$$

$$\delta\tilde{M} = \tilde{U}(x,y,z,t)\left[\tilde{U}_{ref}(x,y,z,t) - \tilde{U}(x,y,z,t)\right] \tag{18}$$

where $U$ is the streamwise velocity in the simulation and $U_{ref}$ is the streamwise velocity in a reference simulation, i.e. a simulation without the wind turbine but with the same inflow and boundary conditions. This operation allows removing the atmospheric shear and low-velocity eddies of the ABL that can be mistaken with the wake. A moving-average operator $\tilde{\cdot}$ is applied on the velocity field with a window of seven frames (i.e. seven seconds). This window size is chosen to smooth the data and facilitate the wake tracking while not impacting significantly the resulting time series of the wake centre's coordinates. The wake outline is then defined as the best fit of $\delta\tilde{U}$ isoline that encloses a surface $S$ such as:

$$\rho \int\int_S \delta\tilde{M}dS = \overline{T} \tag{19}$$

where $\rho$ is the density of the fluid and $\overline{T}$ is the mean thrust. The wake centre is then computed as the velocity deficit centroïd of $S$. An illustration of this algorithm at $x/D = 6$ for an arbitrary time step is given in Fig. 3. This post-processing is performed with the python post-processing tool SAMWICH (Quon et al., 2020) where this algorithm is referenced as *Constant Flux* or *CstFlux*. This algorithm has been chosen for its high success rate and physically-consistent fields in the MFOR (Jézéquel et al., 2022). Finally, several extreme values of $y_c$ and $z_c$ are considered outliers (in the worst case it concerns about 5% of the frames) and manually removed from both the FFOR and MFOR datasets.

## 3.5 Limitations

To compute terms (I) to (VII) of Eqs. 12 and 15, it is needed to start from the unsteady field $U_{MF}$ and apply the Reynolds decomposition and operator $\widehat{\cdot}$ i.e. reverse Eq. 1. To avoid losing any data, one should compute the MFOR on a grid spanning from $y_{min,FF} + \min(y_c)$ to $y_{max,FF} + \max(y_c)$ and similarly in the vertical direction. For strong meandering cases, it would result in a very large grid that would be computationally costly to manipulate. It has thus been decided to restrain the MFOR to $\{y, z\} = \{[-2.5D, +2.5D], [-2D, 2D]\}$. Consequently, some data is missing in the MFOR, leading to unavoidable small differences between the left and right hand-sides of Eqs. 12 and 15.

Given that the ground is located around $z_{FF} \approx -1.2D$, the velocity field $U_{MF}$ at $z_{MF} < -1.2D - z_c(t)$ is undefined since it is located under the ground (the grey region in Fig. 3) so this part of the velocity field is ignored when computing the mean velocity and TKE in the MFOR. Consequently, the statistics (mean and variance) near the ground in the MFOR are computed with fewer samples than those at higher positions.

The wake tracking and the computation of each term of Eqs. 12 and 15 is a costly post-processing, in terms of computational resources and memory. Given the relatively low time step imposed by the ALM it was not feasible to apply this algorithm to every LES time step, so a sampling frequency of 1 Hz has been chosen to store the output velocity field of Meso-NH. This means that all the variations of the wind velocity at frequencies higher than 1 Hz are not taken into account in this work, nor is the subgrid turbulence. Since subgrid turbulence is a prognostic variable in the 1.5-order closure used in Meso-NH, one can compute the ratio between subgrid and total turbulence. It is between 1 % and 5% in the neutral and unstable case but can reach more than 20 % in the stable case. This highlights the difficulty of simulating strongly stratified ABL, but our results have been successfully compared to the SWiFT benchmark (Jézéquel et al., 2021), so they will be used nonetheless.

The error induced by the choice of the 1 Hz sampling has been estimated with a 95 % confidence interval and discussed in Appendix A for the MFOR turbulence and all the terms of Eq. 15. In short, the chosen sampling frequency of $1\,Hz$ is sufficient for the neutral and unstable case. In the stable case, due to the higher share of small-scale turbulence, the 95 % interval is large, indicating that a higher sampling frequency would improve the accuracy of the LES.

Due to numerical limitations, the duration of the simulations was constrained to 80, 40 and 10 min (see Table 1). To estimate the statistical convergence, each of these segments is divided in two and the difference between the two sub-segments and the full-simulation is assessed in Appendix A. It appears that the stable and the neutral simulations would barely benefit from an extension of their duration whereas the unstable simulation seems to change significantly between the two sub-segments of 20 minutes. This is particularly true for terms (V) and (VI) of the turbulence breakdown equation (Eq. 15).

Finally, the streamwise component of the velocity is computed in the following, in both MFOR and FFOR. In all the following, the mean streamwise component of the velocity will be noted $U_x$, and the streamwise turbulence $k_x = \overline{u'u'}$ will be used to differentiate from the total TKE.

## 4    Error induced by neglecting the cross-terms.

Once the Meso-NH simulations are performed, $U_x$ and $k_x$ in the FFOR are directly computed as the mean and variance of the unsteady streamwise velocity field. The wake tracking algorithm described in Sect. 3.4 is applied to get the unsteady streamwise velocity field in the MFOR. The Reynolds decomposition and meandering operator $\widehat{\cdot}$ can then be applied to get the values of terms (I) and (II) of Eq. 12 and terms (III), (IV), (V), (VI) and (VII) of Eq. 15.

The objective of this section is to quantify the importance of each term and to estimate the error induced by neglecting the cross-terms in the velocity and turbulence breakdowns, for instance in the DWM model or in the model developed in the companion paper. The focus is on the neutral case to keep a concise section. The normalised root-mean-square error (RMSE) indicator (Eq. 20) is used to quantify different levels of approximation with the actual results in the FFOR.

$$\text{RMSE} = \sqrt{\frac{\sum_{i=1}^{N}(\alpha - \alpha_p)^2}{N}} / (\alpha_{max} - \alpha_{min}) \tag{20}$$

where $\alpha$ is the reference value (directly extracted from Meso-NH), $\alpha_p$ is the predicted value, $N$ is the number of samples and $\alpha_{max} - \alpha_{min}$ is the range of $\alpha$ over those samples. When the RMSE is computed on a $Y - Z$ plane, only the truncated plane ($y \in [-2D, 2D], z \in [-1D, 1D]$) is used to avoid edge effects and then $N$ denotes the number of mesh points in this plane.

### 4.1    Velocity field

In Eq. 12, the velocity is separated into terms (I) and (II). The vertical profiles of these terms for the neutral case are plotted in Fig. 4 for several downstream positions. The term (I), which is the convolution of the velocity in the MFOR with the distribution of wake centre position, actually fits very well with the velocity in the FFOR. Small differences only appear in the near wake. Term (II) is plotted on a secondary axis (displayed at the top of the figure) to show that it has a negligible value: less than $0.3$ m/s in absolute value, i.e. less than $4\,\%$. In the stable case it is even more negligible but in the unstable case (both are not shown here for conciseness), it takes slightly larger values of about $0.5$ m s$^{-1}$ i.e. $10\,\%$ at the wake centreline in the near wake. As it can be seen at the bottom of the profiles, the main role of this term in the far wake is to reproduce the shear near the ground that is missing in the MFOR, and thus not present in term (I).

From this first observation, it seems acceptable to neglect the term (II). The effect of this assumption can also be measured with a global variable. It has been chosen to investigate the error induced by neglecting the term (II) on the available power, since predicting the power output of a farm is a direct application of analytical models. The available power is here defined as:

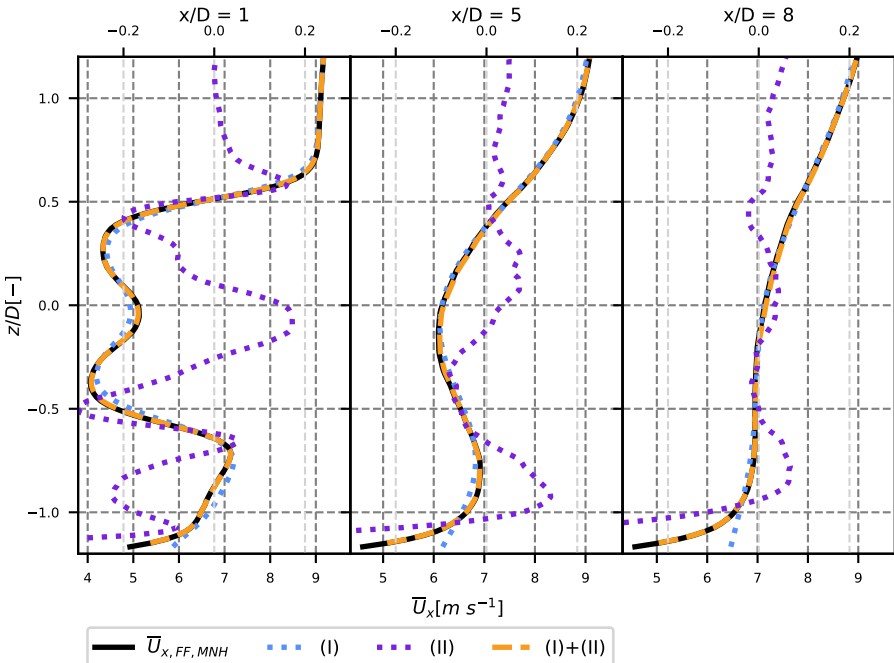

**Figure 4.** Contribution of terms (I) and (II) from Eq. 12 to the velocity in the wake of the neutral case, compared to the velocity in the FFOR. Term (II) is plotted on a different scale (top axis).

$$P_a(x) = \rho \int_S \overline{U}_x^3(x,y,z)dydz \qquad (21)$$

where $S$ is the surface of a virtual wind turbine located at position $x$ behind the wake-emitting turbine, with hub height at the same lateral and vertical position: $y = 0$ and $z = 0$ position. This quantity is computed for (I) and (I)+(II) at each available position downstream of the wind turbine, and compared to the same quantity directly computed on the Meso-NH field in the

290 FFOR $P_{a,FF}$.

From Fig. 5, it appears that neglecting term (II) in the neutral case leads to a slight overestimation of the available power in the near wake of the wind turbine. The estimation is however fairly good, especially for a wind turbine located further than 3D downstream where the overestimation drops below 2 %. The relative error is larger in the unstable case, going from +5 % to -6 % between 1D and 8D downstream. This negative value shows an underestimation of the mean velocity by term (I) in the far

wake. One can note that, at these positions, the tracking algorithm of the unstable case is less reliable, so it could be the source of the error. In such a case, approximating $\overline{U}_{FF}$ with (I) would be correct and the error would come from our methodology. In the stable case, the error is much lower: less than 0.3 %. Further study should be used to determine whether the growing importance of term (II) in the unstable case comes from post-processing errors or an actual physical phenomena. Overall, if the velocity near the ground is not of interest, approximating the FFOR velocity as the term (I) alone as it is done in the DWM can

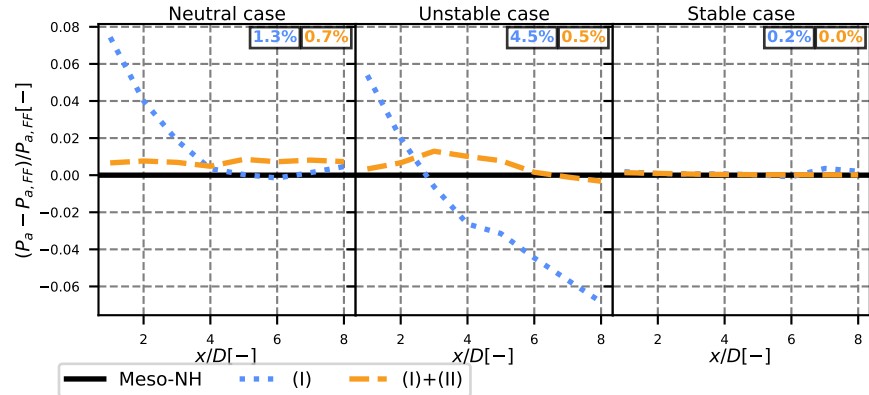

**Figure 5.** Available power predicted by (I) (blue) and (I)+(II) (yellow) for the three simulation cases. The value is normalised with the results in the FFOR (black line). The RMSE of $P_a$ averaged over all the $x$ positions is displayed at the top in the corresponding color.

thus be acceptable given the low error on estimated power. This is especially relevant since the term (II) seems very chaotic (see Fig. 4) and thus hard to model.

### 4.2 Turbulence field

The same study is performed for the turbulence field in the wind turbine wake. The vertical turbulence profiles for the neutral case are plotted for different levels of approximation, at different positions downstream in Fig. 6. In the DWM model, only the meandering (III) and rotor-added turbulence (IV) terms are retained. This corresponds to the blue curve: despite an overall good order of magnitude, it can be seen that the vertical asymmetry is not sufficiently pronounced, leading to an underestimated value of $k_x$ at the top tip and overestimated value at the bottom tip. This issue, especially true in the near wake, has already been observed in another work that used an equation similar to Eq. 15 (Conti et al., 2021) to compare the DWM results to *in situ* measurements. If horizontal profiles at hub height are plotted instead (shown in the companion paper), the results are much better and the DWM approximation seems suitable, for the neutral but also the unstable and stable cases.

Adding the covariance term (V) along with terms (III) and (IV) (purple curve in Fig. 6) corrects for most of the vertical asymmetry of the turbulence profiles and leads to a rather good estimation of the maximum turbulence values at the top and bottom tips. The main effect of adding term (VI) (red curve) is to take the spatial small-scale variations into account, bringing the total $k_x$ even closer to its reference value. As pointed out previously, the term (VII) is the square of the term (II): like the latter, it mainly has an effect near the ground but is otherwise negligible.

To quantify more clearly these differences, the maximum axial turbulence $k_x^M(x)$ is studied. It is computed directly in the FFOR ($k_{x,LES}^M(x)$) and for different levels of approximation from Eq. 15. Their evolution with the downstream distance is plotted in Fig. 7, normalised by $k_{x,LES}^M(x)$ and the same colour convention as in Fig. 6 is used.

In the neutral case, neglecting the cross-terms leads to an underestimation of about 6 % to 12 % of the maximum turbulence in the wake. In the far wake (beyond $x/D = 5$) the error decreases but this is a numerical artefact: due to edge effects, large

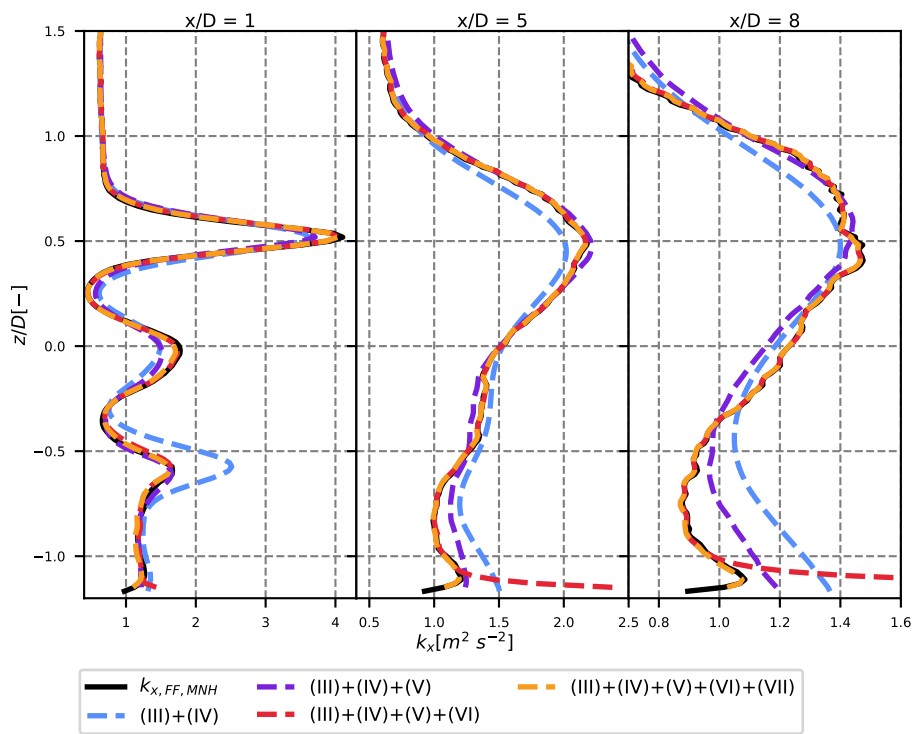

**Figure 6.** Streamwise turbulence profiles in the wake of the wind turbine for different levels of approximation. Neutral case at three positions downstream (1, 5 and 8 diameters).

TKE values are observed near the ground, and thus the maximum TKE is detected at this location instead of at the top tip. Adding the covariance term (V) allows to bring this number down between 2 % and 6 %, and adding term (VI) to this total leads to a negligible underestimation (around 1 %). Term (VII) has a negligible effect on the maximum turbulence (orange and red curves are superimposed). The remaining gap is attributed to the error reconstruction due to the MFOR not being large enough (see Sect. 3.5).

For the unstable case, the same orders of magnitude are observed for the different $k_x$ approximations: adding the convolution term (V) reduces the relative underestimation of $k_x^M$ by at least half and using (III)+(IV)+(V)+(VI) leads to a fairly good approximation. In the stable case, however, the error remains between 5 and 10 %, independently of the level of approximation. This is because despite an improvement over the whole profile, the maximum is not well captured (figure not shown here for brevity). The relatively high error percentage is attributed to low absolute values. Indeed, the error is of the order of magnitude of $0.04 \ m^2 s^{-2}$, which is in the end similar to the other cases. Term (VII) is negligible in every case.

It has been shown in this section that neglecting term (II) as in the DWM model or in the companion paper leads to a rather accurate velocity deficit in the wake and a reasonable estimation of the available power (less than 2% overestimation) for a wind turbine inside the wake, as long as it is positioned beyond $x/D = 3$. For the turbulence breakdown, the term (VII) is also

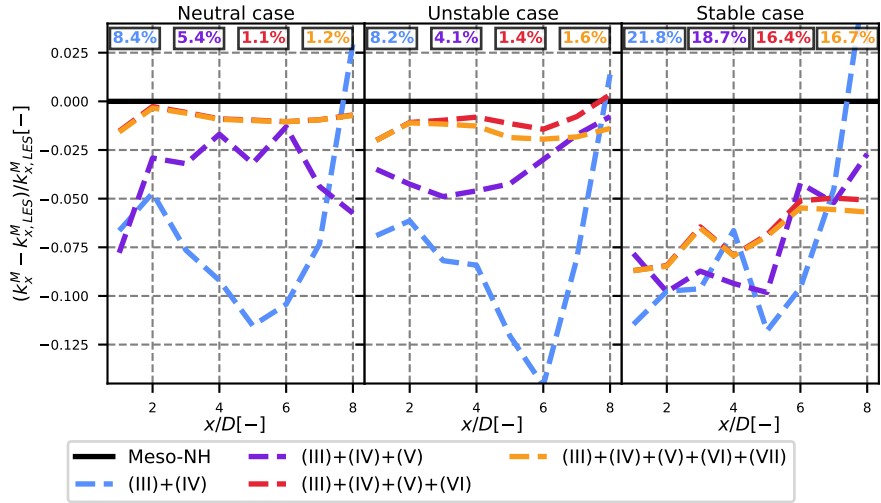

**Figure 7.** Normalised maximum turbulence in the wake for different levels of approximation as a function of the downstream position for the three simulation cases. The RMSE of $k_x^M$ averaged over all the $x$ positions is displayed at the top in the corresponding color.

negligible, but the vertical turbulence profiles are prone to errors when the term (VI) and more importantly the term (V) are not taken into account, leading to an underestimation of the maximum turbulence in the wake. It is now needed to compare the shapes and the relative magnitude of these terms before modelling them.

## 5  Analysis and interpretation of the turbulence breakdown

In this section, the turbulence fields in the wake of the wind turbine are compared for the three cases of stability. The influence of atmospheric stability on each term of Eq. 15 is highlighted and the shape of these terms in the Y-Z plane is analysed.

### 5.1  Shape and values of the terms

The values of each term of Eq. 15 at different Y-Z planes downstream of the turbine in the FFOR are displayed in Figs. 8, 9 and 10 for the neutral, unstable and stable cases respectively. The terms are normalised by the maximum total turbulence in the FFOR $k_{x,LES}^M(x)$ in the 2D plane, so the scale is approximately the fraction of the total axial turbulence represented by each term. Term (IV) contains both the rotor-added turbulence and the inflow turbulence, which is removed by subtracting the reference turbulence field in the MFOR $k_{x,ref,MF} = \overline{U_{x,ref,MF}'^2}(x,y,z)$ taken from the reference simulation (the same simulation without the turbine, see Sect. 3.4) at the same location than the turbulence field with the wind turbine. In the MFOR the rotor-added axial turbulence is thus defined as the difference of axial turbulence between the simulation with and without the wind turbine:

$$\Delta k_{x,MF}(x,y,z) = \overline{U_{x,MF}'^2}(x,y,z) - \overline{U_{x,ref,MF}'^2}(x,y,z) \tag{22}$$

Note that the $y_c(t)$ and $z_c(t)$ computed in the simulation with a turbine are re-used to compute the reference MFOR field and to apply operator $\hat{\cdot}$ to the reference data. The rotor added turbulence can then be defined in the FFOR as:

$$\Delta(\text{IV}) = \overline{\widehat{U_{MF}'^2}} - \overline{\widehat{U_{ref,MF}'^2}} = (\text{IV}) - \overline{\widehat{k_{x,ref,MF}}} \tag{23}$$

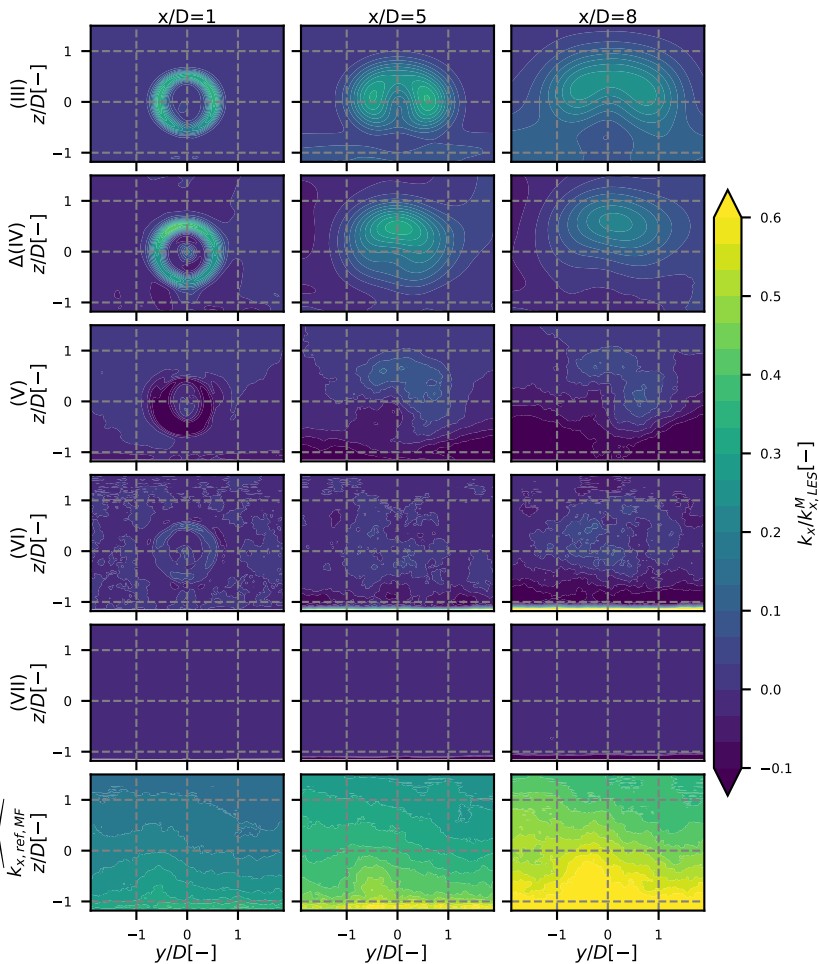

**Figure 8.** 2D maps of the different terms in Eq. 15 for the neutral case. The different rows stand for the different terms and each column is a different position: 1, 5 and 8 diameters downstream. The values are scaled by the maximum TKE in the FFOR at the given $x$ position.

For the neutral case of stability (Fig. 8), the meandering (III) and rotor-added $\Delta$(IV) terms have similar orders of magnitude and contain most of the total wake added turbulence. However, the covariance term (V) cannot be ignored as it rebalances the

total turbulence of about $\pm 10\%$ between the top and bottom regions of the wake, as it has been seen in Fig. 6. Term (VI) also shows non-negligible values, in particular in the far wake where it progressively takes values closer to the other terms, but the shape of this term seems to be randomly distributed (contrarily to term (V) which is located in the rotor-swept area). As stated in Sect. 4, the term (VII) is negligible, except near the ground.

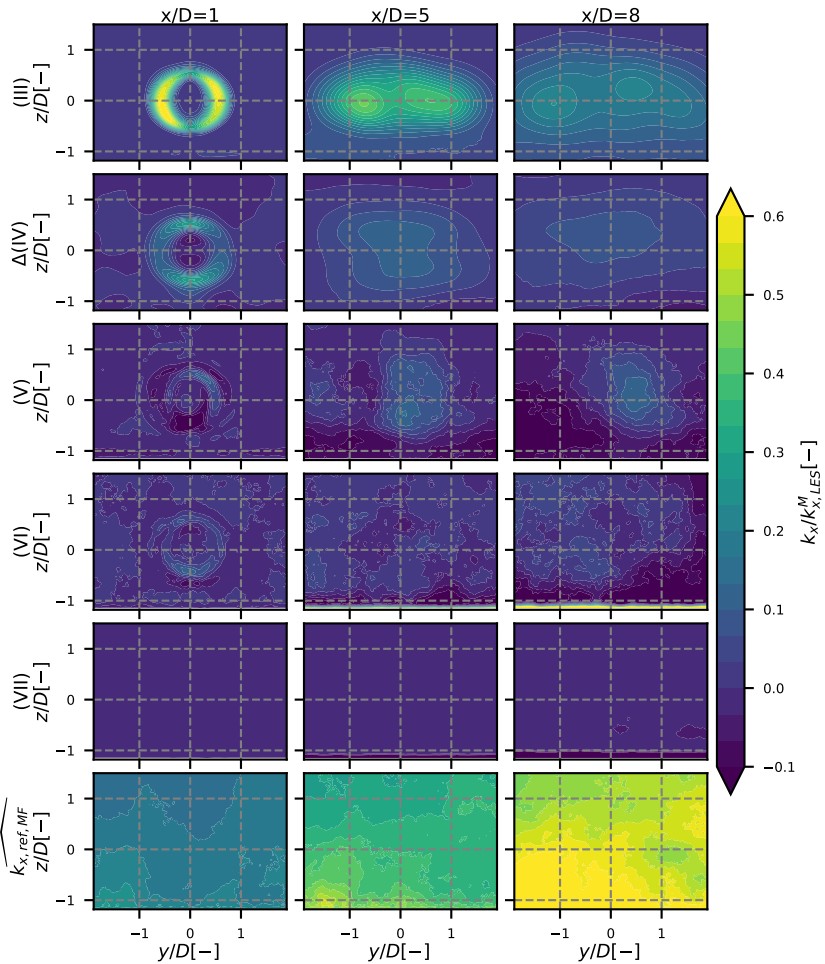

**Figure 9.** Same as Fig. 8 for the unstable case.

Figure 9 has been plotted similarly to Fig. 8 with the results of the unstable case. The meandering term (III) is dominant over the others and the wake is quickly dissipating. The rotor-added turbulence has lower relative values and is more spread than in the neutral case. This is due to larger meandering in the unstable case i.e. a PDF $f_c$ with larger values at the edge and thus more spreading caused by the operator $\widehat{\cdot}$. The covariance term is also not negligible: here it takes values between terms $\Delta$(IV) and (III) in the far wake. In this case, the term (V) is symmetric about the vertical axis instead of the horizontal one. Term (VI) shows lower values, that seem to be randomly distributed as in the neutral case. Term (VII) is still negligible.

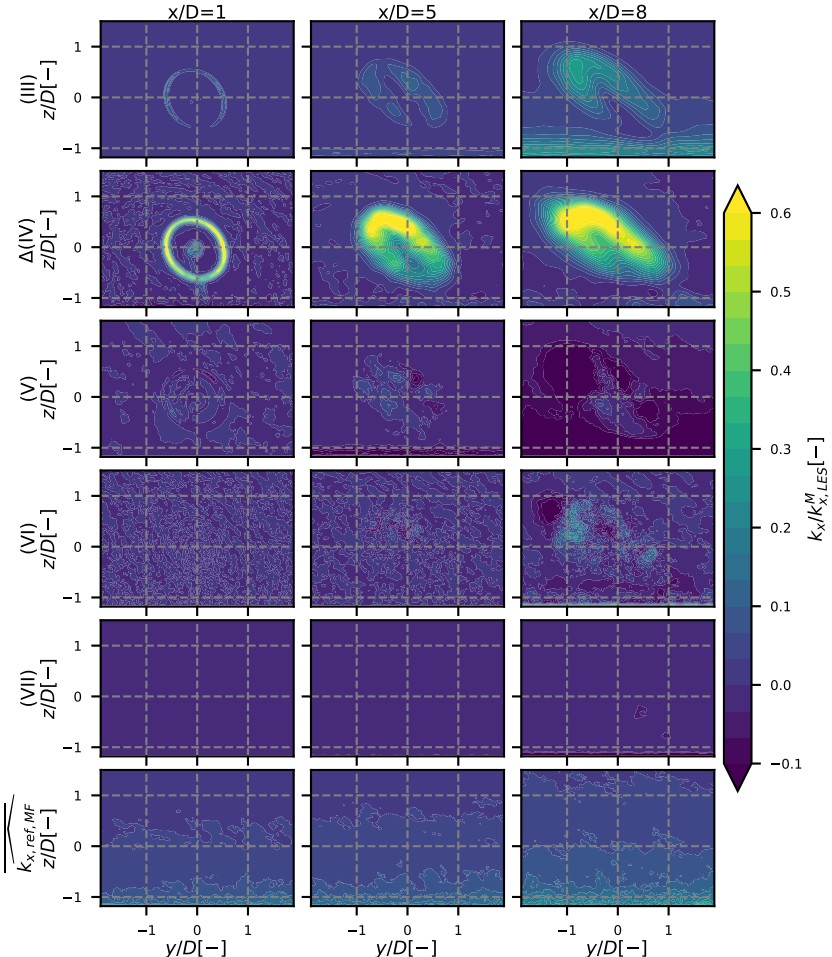

**Figure 10.** Same as Fig. 8 for the stable case.

In the stable case (Fig. 10), it is the rotor-added turbulence that is largely predominant over the meandering and even the upstream terms. This can be explained by the fact that meandering is very weak, so the term (III) is low, the term (IV) is almost not spread by the convolution with $f_c$, and the wake is barely dissipated, even at $x/D = 8$. The covariance term is here negligible except at $x/D = 8$ where it slightly reduces the peak of turbulence at the top-left end of the wake. Term (VI) and particularly term (VII) are negligible in front of the term (IV). The shape of all these terms is skewed due to the strong veer present in the stable ABL.

The reference turbulence in the FFOR $\overline{\widehat{k_{x,ref,MF}}}$ is also plotted in the last line of Figs. 8, 9 and 10 to quantify how the wake turbulence is going back to its unperturbed value: the closer $\overline{\widehat{k_{x,ref,MF}}}$ is to 1, the more dissipated is the wake. At $x/D = 8$ in the unstable case (Fig. 9), the reference turbulence in the FFOR represents the main part of the total turbulence which means

that the wake is almost dissipated. Conversely, in the stable case (Fig. 10), the reference turbulence in the FFOR is negligible,
compared to the other terms, showing that the wake is much less dissipated than in the other cases.

For all cases, the non-zero values of each term in the near wake (first column of every figure) are mostly distributed around
the tip of the blades. For pure-terms (III) and (IV), they are spatially smoothly distributed at $x/D = 5$ and $x/D = 8$. For cross-
terms (V) and (VI) and (VII), the non-zero values at these positions are chaotically distributed spatially and thus harder to
interpret due to a lot of small-scale variations. A statistical averaging of every term over several simulations could provide data
with better spatial coherence and the different terms would thus be easier to interpret. To do so, longer simulations with similar
mean upstream conditions are needed.

## 5.2 Physical interpretation

Term (III) or $k_m$ is the pure meandering term. For a fixed point downstream of the turbine, the meandering of the wake induces
an alternation between low velocity (when the point is inside the wake) and high velocity (when it is outside the wake), i.e.
variance in the unsteady velocity field, which is the definition of turbulence. $k_m$ thus increases with the velocity deficit in the
MFOR and with the amount of meandering. The former decreases with $x$ whereas the latter increases with $x$, often linearly
(Keck et al., 2013a; Ning and Wan, 2019; Brugger et al., 2022). These two contradictory trends lead $k_m$ to be strong and
very localised at the tip of the blades in the near wake and to be progressively smeared as the wake travels downstream. Since
the meandering is stronger in the horizontal direction than in the vertical direction and the velocity deficit is approximately
axisymmetric (see the companion paper for more details), the highest values of $k_m$ in the horizontal plane are stronger than in
the vertical plane.

At a fixed $x$, the maximum values of $k_m$ are localised near the tip of the blades in the near wake and are gradually spread as
the wake travels downstream. The maximum added TI induced by term (III) (i.e. square-root of the maximum value, normalised
by the upstream velocity at hub height) is plotted in dashed lines as a function of $x/D$ in Fig. 11. As seen in Figs. 8, 9 and
10, the meandering-induced turbulence is inversely related to the atmospheric stability, but this term also decreases faster in
the unstable case, likely because the stronger the meandering, the more dissipated is the wake. Consequently, at $x/D = 8$ the
unstable and neutral added TI due to the meandering are almost identical, and the curves would probably switch at larger $x$. In
the stable case, the velocity profile is barely dissipated up to $x/D = 8$ and the meandering starts to take consequent values at
$x/D = 5$, which results in an increase of the added turbulence due to meandering starting from $x/D = 5$. One can predict that
beyond $x/D = 8$, a maximum value is reached, followed by a shape similar to the unstable and neutral case.

Term $\Delta$(IV) noted $k_a$ for 'rotor added turbulence' is the turbulence that would exist in the wake of the turbine if there was
no meandering. This turbulence is mainly due to the velocity gradient in the MFOR, localised at the edge of the wake. It is
affected by the shear of the ABL, leading to a stronger gradient near the top tip and thus stronger rotor-added turbulence. This
is particularly visible in the neutral and stable cases, where the atmospheric shear is significant. Similarly to the velocity field,
this added turbulence is spread by meandering, more strongly in the lateral direction than in the vertical one, leading in the
unstable case to lower values of $k_a$ at the side tips than the bottom tip despite the atmospheric shear being stronger at the side.
This spreading of meandering also induces lower values of maximum added turbulence for lower stability cases (dotted lines

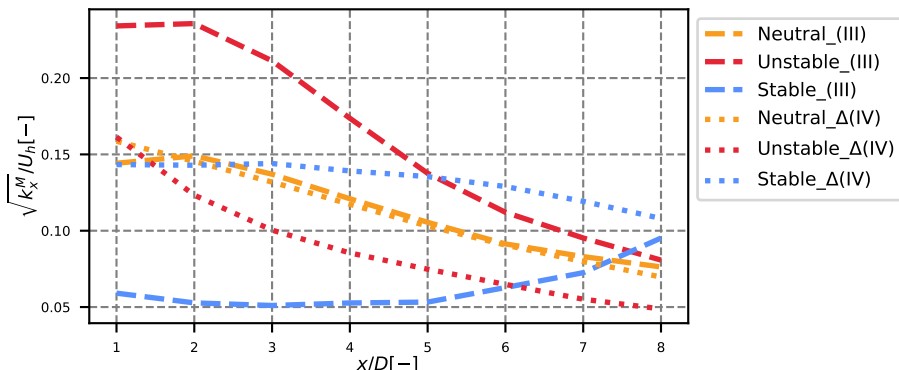

**Figure 11.** Evolution of the maximum value of terms (III) and $\Delta$(IV) with $x$, normalized by the velocity at hub height.

in Fig. 11). To analyse the shape of the rotor-added turbulence before the spreading due to meandering, one needs to look at
410 the values of $k_a$ in the MFOR. It is normalised with the hub height velocity to give the added TI in the MFOR:

$$\Delta TI_{MF} = \frac{|\Delta k_{x,MF}|}{\Delta k_{x,MF}} \cdot \frac{\sqrt{|\Delta k_{x,MF}|}}{U_h} \tag{24}$$

Equation 24 allows identifying in which region the turbulence in the MFOR is lower than the unperturbed turbulence, without leading to undefined values of the square root. The values of $\Delta TI_{MF}$ in the three cases of stability are plotted in Fig. 12 at different positions downstream.

Despite strongly different values of $\Delta$(IV) among the different cases, Fig. 12 show that the atmospheric stability mostly affects meandering and not the field in the MFOR. Indeed, the magnitude of the normalised added TI in the vicinity of the turbine (at x/D = 1) is about 19 % in the neutral and unstable case, and the slightly lower value in the stable case (about 15 %) is attributed to smaller integral length scales upstream the turbine. These discrepancies are small compared to those observed in the FFOR, where the added TI reaches about 22%, 27 % and 16% for the neutral, unstable and stable case (Jézéquel et al.,
2022). The skewed shape of the stable case is attributed to the veer that appears in such ABL, but is negligible in neutral and unstable ABL. As the wake travels downstream, the asymmetry increases, in particular for the neutral and stable cases, but the magnitudes of $\Delta TI$ are still similar among the different cases despite different values of atmospheric stability, shear and hub height velocity. The asymmetry is attributed to the ambient shear, which increases with atmospheric stability. Negative values of $\Delta k_{MF}$ are observed in the near wake between the wake centre and the edge in the neutral and unstable cases and also in the
bottom of the far wake in the neutral case. This indicates a transfer of energy from such regions to the high turbulence region, i.e. the edge and the top of the wake. Overall, this figure shows that the different values of $\Delta$(IV) among the cases mainly come from the meandering operation and only slightly from the MFOR turbulence itself.

The value of the cross-terms (V), (VI) and (VII) is 0 either if there is no meandering (i.e. $\widehat{a} = a \,\forall a$) or if there is no turbulence in the MFOR ($U'_{MF} = 0$). Even though the latter can be assumed in some models, none of these conditions is fulfilled in real
cases. It has been chosen to regroup the two terms $\overline{\widehat{U_{MF}U'_{MF}}} - \overline{\widehat{U_{MF}}\widehat{U'_{MF}}}$ into one single covariance term (V) since those two

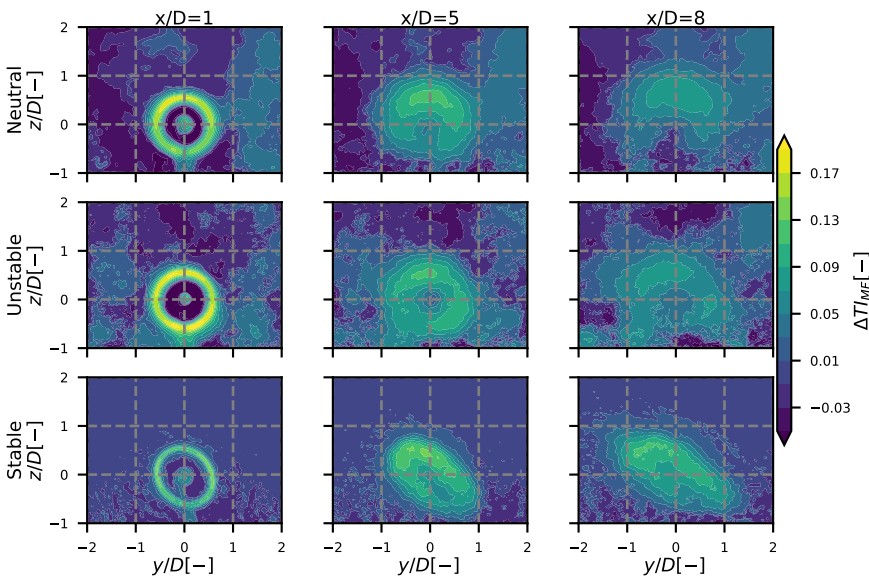

**Figure 12.** 2D map of the added turbulence in the MFOR, normalised by the velocity at hub height.

terms were very large (in absolute value), compensating each other, and thus hard to interpret. Mathematically, this covariance term quantifies how the mean and varying parts of $U_{MF}$ evolve together once displaced by the meandering operation $\widehat{\cdot}$. In the near wake, the non-zero values are distributed at the tip of the blades and then gradually expand in the whole wake. Negative and positive values are symmetrically distributed (along the horizontal and vertical axis for the neutral and unstable cases respectively). From these results, no physical interpretation nor a relation between the values of $U_x$ or $k_x$ in the wake with the term (V) could be found yet. Modelling the covariance term has thus not been achieved in the companion paper, but the authors are confident that it is an important step toward improvements of wake models based on the meandering. If more data were available, one could perform an ensemble average and hopefully find a shape easier to interpret for this term.

Term (VI) can be viewed as the varying part of turbulence: before being moved by the meandering and averaged, this term is the varying part of the square of the deviation from the mean (in opposition to $k_{x,MF}$ which is the mean part of the square of the deviation from the mean). In the near wake, positive values are present at the tip of the blades in the neutral and unstable cases, but also outside of the wake. It then gradually expands in the whole wake and seems randomly distributed in the wake region with negative and positive values. From Figs 8, 9 and 10, it seems that excepted systematic negative values near the ground ($z < -0.5D$), this term mainly reproduces the spatial non-homogeneity of the wake and is thus not vital to be represented in an analytical model.

Term (VII) is always negative from its mathematical formulation: similarly to the viscous dissipation in the Navier-Stokes equations, it is a sink of energy. It has negligible values in all the stability cases. This last result should be taken with care: if the analogy with the viscous dissipation hold for this term, it means that it concerns small scales eddies, i.e. variations of the

wind velocity at high frequency. Yet, as explained in Sect. 3, only the variations of time scale larger than 1 s are captured with the post-processing used in this work because of memory limitations. With a sampling frequency higher than 1 Hz, this term may have higher values.

It is important to note that all these results are sensitive to the wake tracking method: despite that the method used here being among the most reliable available in the literature, there are always frames where the tracking failed, plus the limitations described in Sect. 3.5. For instance, the turbulence field in the MFOR (see Fig. 12) is noisier and noisier as the wake travels downstream and in particular in the unstable case, which can be interpreted as a consequence of the tracking algorithm being less and less reliable. This remark can be extended to all the terms of the turbulence equation presented in Figs. 8, 9 and 10. Moreover, the values and shapes of the different terms (in particular the cross-terms) might also change depending on the turbulence field, i.e. the eddies of the ABL, even for similar mean atmospheric conditions.

## 6 Conclusions and perspectives

In models predicting wake meandering such as the DWM, it is assumed that the turbulence in the wake can be separated into two parts: the turbulence generated by the rotor and the turbulence generated by meandering. In this work, the turbulence in the FFOR has been developed as a function of the two terms aforementioned and it appears that three cross-terms are missing, thus implicitly neglected in DWM-type models. A similar conclusion is drawn for the velocity, with one missing term.

To quantify the importance of each of these terms, and estimate the error induced by the assumptions of such models, LESs with an actuator line are performed to model the wake of an isolated wind turbine inside an ABL. The modelled turbine is the modified Vestas V27 used in the SWiFT campaign of measurements, and three cases of atmospheric stability are investigated: near-neutral, unstable and strongly stable. The instantaneous wake centre is detected at different planes downstream of the turbine (from 1D to 8D) to compute the velocity field in the MFOR. The main conclusions are the following:

 – Neglecting the cross-term of the mean velocity equation leads to small differences in the computation of the mean velocity profile in the FFOR. For the neutral case, the corresponding error leads to a less than 2 % overestimation of the available power in the wake of the wind turbine for a turbine located further than 3 D behind the wake emitting rotor.

 – Neglecting cross-terms in the computation of turbulence in the FFOR leads to vertical profiles where the imbalance between the turbulence at the bottom tip and the top tip is underestimated. Adding the three missing cross-term allows to correct this error and reduce the overall RMSE.

 – In the unstable case, the meandering term is dominating the total streamwise turbulence whereas in the stable case, it is the turbulence added by the rotor which is dominant. In the neutral case, these two terms are of similar magnitude and overall larger than the cross-terms. These cross-terms, especially the so-called covariance term however show local values sufficiently strong to correct significantly the maximum axial turbulence in the wake.

The statistical convergence of the data has been assessed and showed that increasing the sampling frequency would most likely improve the reliability of the stable case but would have few effect for the two other cases. On the other hand, increasing

the simulation time would probably change the unstable results but have low impact for the other cases. The uncertainty is the highest on the cross-terms of the turbulence breakdown equation, but the pure-terms are subject to only small uncertainty. For a better interpretation of these terms, it may be important to perform ensemble simulations to get more reliable fields.

It must be noted that these conclusions are drawn on the results of three particular cases of atmospheric stability and one model of turbine that can be regarded as rather small compared to modern rotors. The orders of magnitude given in this work should not be considered universal but are a good indication that for an accurate version of DWM-type models, the cross-terms (or at least the covariance term) must be taken into account. In the companion paper, an analytical model for the dominant terms is developed on the neutral and unstable cases presented herein.

### Appendix A: Statistical convergence of the results

In this appendix, an analysis of the statistical convergence of the three simulations is proposed to give a glimpse to the reader of the uncertainty of the presented data. Indeed, the quality of our data is limited in two ways:

– First, a sampling frequency of 1 Hz has been set for the three simulations. This value has been chosen accordingly to the SWiFT benchmark (Doubrawa et al., 2020), from which the present simulations are inspired. However, one can argue that it does not capture sufficiently the small-scale variations of the signal.

– Moreover, a length of 80 minutes was initially the target for the three simulations. However, the unstable simulation is computationally expensive and the stable simulation diverges to unrealistic results if it runs for too long due to the strong negative heat flux. Consequently, the segment length was reduced to 40 and 10 minutes for the unstable and stable cases, respectively. Even for the 80 minutes though, it is not sure *a priori* that the simulation has run for long enough to have statistically converged results.

These two points have been investigated independently with two different methods in this appendix. They are used to quantify the uncertainty on every term of Eq. 15 and on $k_{x,MF}$, for the three simulation cases. Figures A1, A2, A3 and A4 thus allow to respectively estimate the uncertainty of the quantities plotted in Figs. 8, 9, 10 and 12 for the estimation of the corresponding uncertainty. Obviously, this statistical convergence study can be expanded to other quantities but the aforementioned are considered to be the core of the present work.

### A1  Sampling frequency

In the most refined domain, the Meso-NH code runs at a time step close to the millisecond (see Table 1). Thus, estimating the statistics (mean, variance) of the wind velocity from a signal sampled at a frequency of 1 Hz is equivalent to estimating statistics on a reduced population. In our case, the sample time is much greater than 1 s (at least 600 s in the stable case), but much lower than the actual time-step of the signal (which is about 1 ms, depending on the Meso-NH time-step). In such case, the law of large numbers indicates that the mean computed on the reduced size population is a random variable which follows

a normal law. The 95 % confidence interval of the mean of a variable $X$ computed from a population of reduced size can thus be computed as

$$CI(X) = \frac{\sigma_X}{\sqrt{n}} \tag{A1}$$

where $\sigma_X$ is the standard deviation of the variable $X$ and $n$ is the sample size. The turbulence and every term of Eq. 15 can be written as the mean of a given variable, except for term (VII). Consequently, Eq. A1 is applied on the time serie corresponding to each of these terms, before the time averaging is applied. For instance, for term (III), it gives:

$$\sigma_{(III)} = \sqrt{\mathrm{var}\left(\left(\widehat{\overline{U_{MF}}} - \overline{\widehat{\overline{U_{MF}}}}\right)^2\right)} \tag{A2}$$

For term (VII), it is assumed that the confidence interval is equal to the square of the confidence interval of term (II). This may be a strong approximation, but this term is anyway found to be negligible, and thus plays only a small role in the total turbulence budget.

The confidence interval is plotted in the shaded area in Figs. A1, A2, A3 and A4. It appears that there is a strong confidence in our results for the pure-terms (terms (III) and (IV)) of the turbulence breakdown equation (Figs. A1, A2 and A3). The cross-terms, in particular term (VII) in the stable case, seem to be less reliable but the results can still be considered satisfying. However, the confidence interval is more important for the MFOR turbulence, in particular for the stable case (Fig. A4).

This confidence interval quantifies the uncertainty towards the small-scale turbulence. It is thus not surprising that it gets higher values in the stable case (where turbulence is mostly of small scale) and in particular for the turbulence in the MFOR and for the term (VI) which is attributed to the spatial non-homogeneity, and thus to the small-scale variation. A higher sampling frequency for the stable case would seemingly improve the reliability of the simulation, but for the two other cases, the sampling at 1 Hz seems suitable.

## A2   Length of the simulations

The length of the simulations (80, 40 and 10 minutes for the neutral, unstable and stable cases) has been arbitrarily chosen depending on numerical constraints. Even though some works in the literature are based on 10-minute-long simulations, it should be ensured that this choice is sufficient to take into account all the large-scale variations of the ABL.

To do so, the velocity time series are separated into two equal sub-segments (thus of length 40, 20 and 5 minutes for the neutral, unstable and stable cases). The whole post-process is reproduced on these sub-segments, and the resulting profiles are plotted in dashed black lines in Figs. A1, A2, A3 and A4. One can argue that if the sub-segments are similar to the full simulation, increasing further the computational time would have likely no effect, whereas if significant differences are observed, the simulation is probably not well converged. Note that it is normal if the mean of the two sub-segments (dashed black line) is not exactly equal to the full segment (continuous colored line) because statistics of higher order than averages are at stake. For instance, the total variance is always larger than the mean of the sub-segments variables.

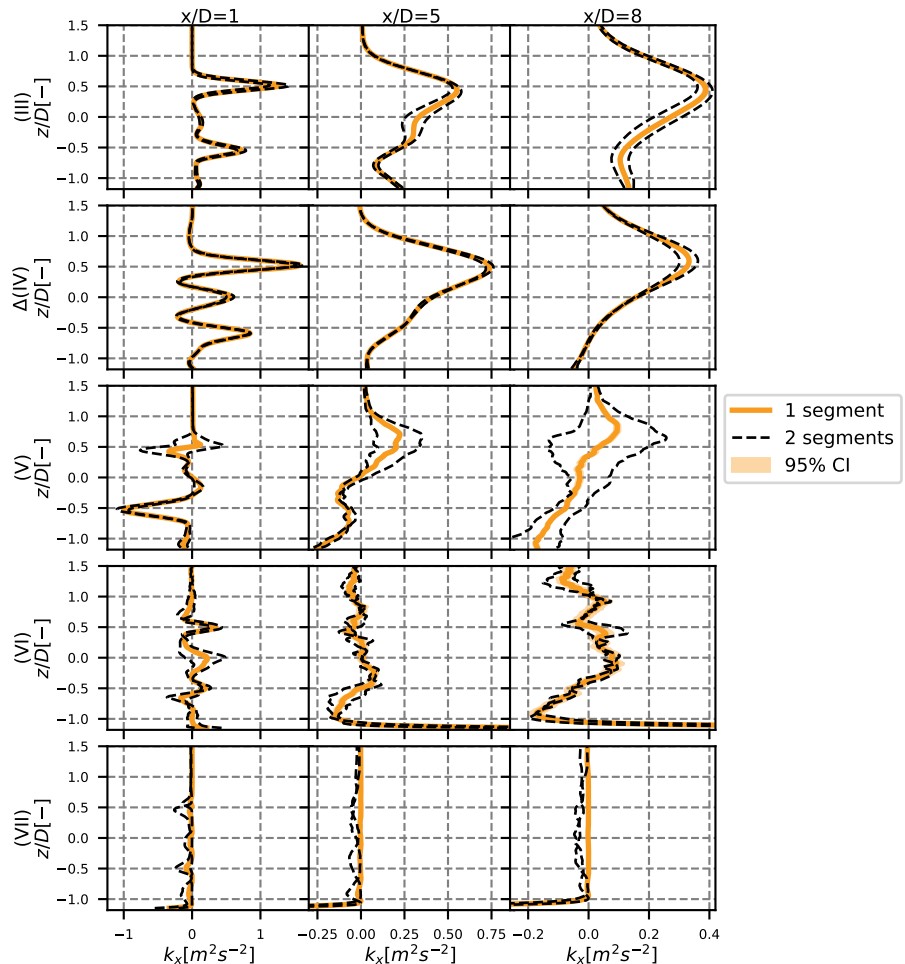

**Figure A1.** Vertical profiles at $y = 0$ of the different terms of the turbulence breakdown equation, for the neutral case. The 95 % confidence area is plotted with a shaded area and the results with two sub-segments in dotted black lines. The different rows stand for the different terms and each column is a different position: 1, 5 and 8 diameters downstream.

This concept of convergence should however be taken with care here because the ABL is in permanent evolution due to the constant heat flux at the ground that changes progressively its characteristics. This is particularly true for the unstable case where the mean wind direction is not equal among the two sub-segments and where the atmospheric conditions will likely never reach a quasi-steady state.

Conversely to the sampling frequency, this method measures the uncertainty of the large-scale turbulence. It is thus not surprising to see that the uncertainty is higher in the unstable case (Fig. A2). Again, the uncertainty is higher on the cross-terms, and in particular on the term (VI). For the turbulence in the MFOR, the uncertainty due to the length of the simulations is

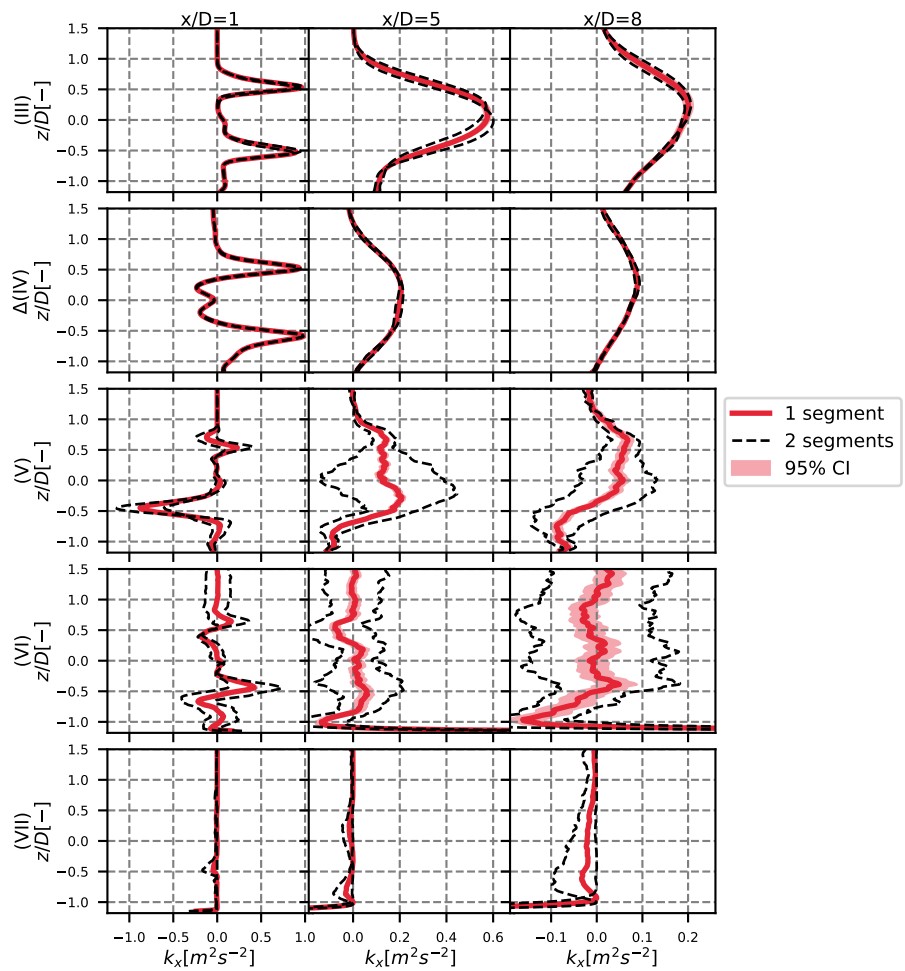

**Figure A2.** Vertical profiles at $y = 0$ of the different terms of the turbulence breakdown equation, for the unstable case. The 95 % confidence area is plotted with a shaded area and the results with two sub-segments in dotted black lines. The different rows stand for the different terms and each column is a different position: 1, 5 and 8 diameters downstream.

limited, which is consistent with the fact that there are supposedly only small-scale variations in this frame and thus, increasing the segment length would have likely no effect.

550 As a conclusion, a higher sampling frequency could give more reliable results in the stable case, and a longer simulation time may have been needed for the unstable case. This is particularly true for the term (VI) which has the largest level of uncertainty. Nevertheless, the uncertainty on the other terms and the streamwise turbulence in the MFOR seem sufficiently low to maintain the conclusions drawn at the core of this article.

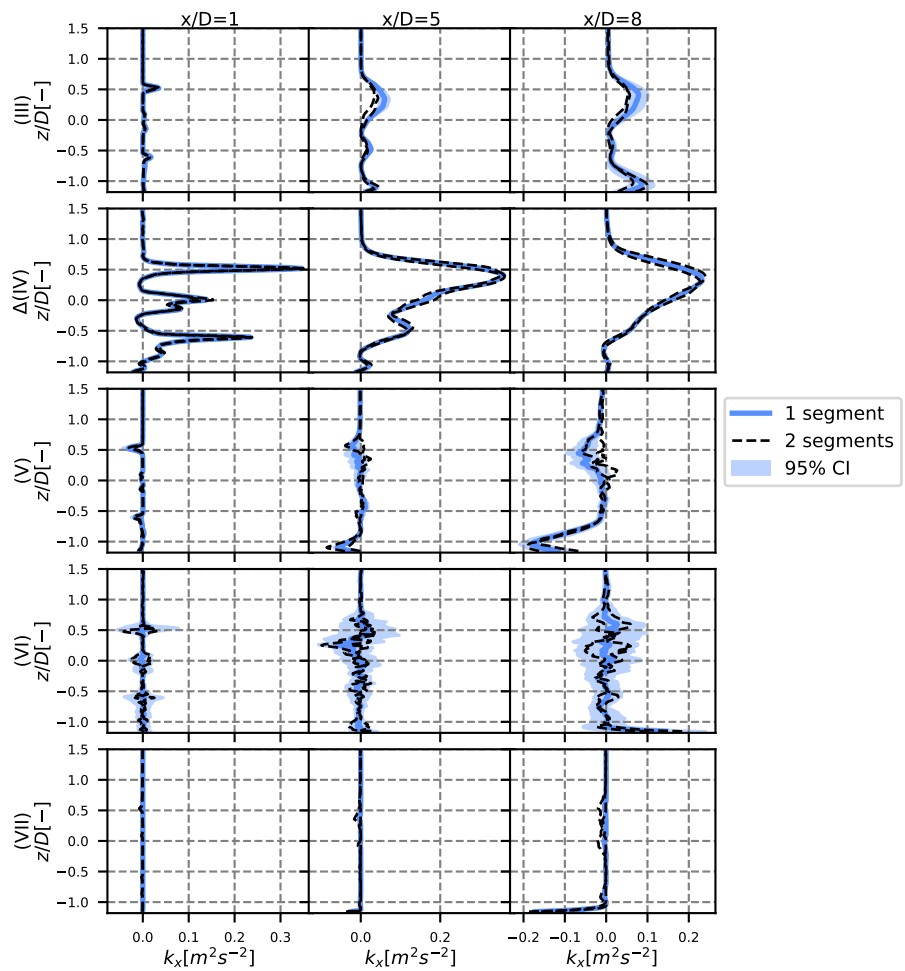

**Figure A3.** Vertical profiles at $y = 0$ of the different terms of the turbulence breakdown equation, for the stable case. The 95 % confidence area is plotted with a shaded area and the results with two sub-segments in dotted black lines. The different rows stand for the different terms and each column is a different position: 1, 5 and 8 diameters downstream.

*Code and data availability.*  The code Meso-NH is open-source and can be downloaded on the dedicated website. The authors can provide the source code of the modified version 5-4-3 that was used in this work. The results of the LES simulations can also be directly provided.

*Author contributions.*  EJ developed the equations, and performed the simulations with VM. All the authors worked on the interpretation of the results. The manuscript has been written by EJ with the feedbacks of FB and VM. The data used for the plot presented here and in part 1 are available under this online deposit: 10.5281/zenodo.6562720.

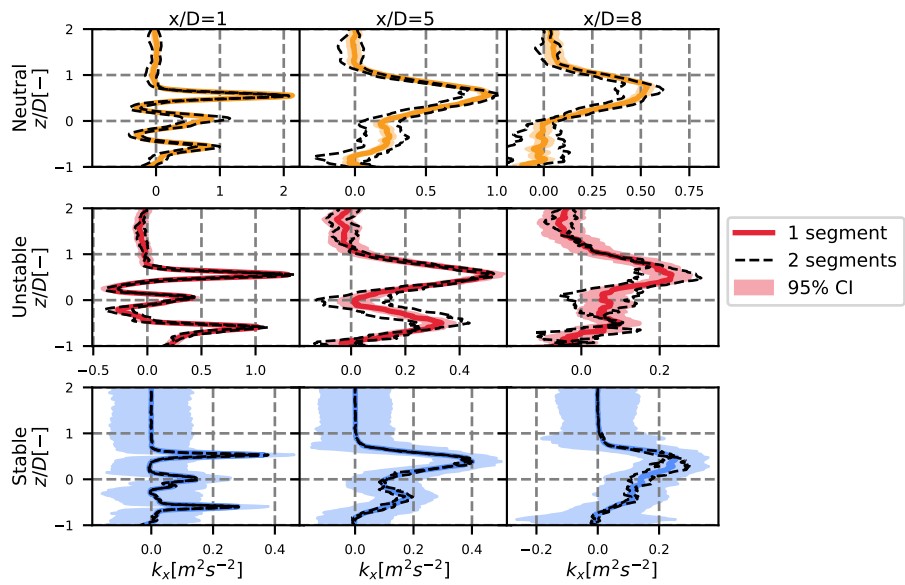

**Figure A4.** Vertical profiles at $y = 0$ of the streamwise turbulence in the MFOR. The 95 % confidence area is plotted with a shaded area and the results with two sub-segments in dotted black lines. The different rows stand for the different cases and each column is a different position: 1, 5 and 8 diameters downstream.

*Competing interests.* The authors declare that they have no competing interests.

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
