# Peer review of "Breakdown of the velocity and turbulence in the wake of a wind turbine - Part 1: large eddy simulations study."

_Wind Energy Science, 2022_

## Author Comment (AC1)

Dear editor and reviewers, thank you for taking the time to examine our work. Several weak points were pointed out, which we hope to have comprehensively answered in the final version. Please find our answer (in bold blue) to the reviewers comments (in black).

**RC1:**

In the manuscript the authors study the development of a wind turbine wake is studied in a moving reference frame. To be specific, the mean velocity and turbulence in the fixed field of reference are broken down into different terms, which are analyzed in the moving reference frame. In the present work the authors show that including the cross-terms improves the results. The effect of the cross terms is relatively modest given that analytical wake models generally have various simplifications. It is argued that this analysis is relevant to extend models like the DWM, which is submitted as a second paper, but not discussed here. Overall, such modeling efforts are relevant for the community to get improved insights. In any case the presented analysis does provide additional insights into the development of a wake. Although the statistical convergence of the data seems limited and uncertainty analysis is limited. These aspects should be improved.

- Line 228-230: "are not taken into account in this work, nor is the sub-grid turbulence. The latter is negligible in the unstable and neutral cases but can reach more than 10% in the stable case." --> How did you determine the sub-grid turbulence, and what day you mean by negligible? That seems a bit of a strong statement.

**A 1.5-order turbulence closure is used in Meso-NH. An equation for subgrid turbulence is resolved, which gives a value that can be compared with the resolved turbulence. See the 6$^{th}$ comment of RC2 for more details.**

- Line 314-315: "I wonder how accurate that is". How reproduce-able are your simulations? The turbulence in turbulence simulations is not necessarily exactly reproducible. The statistics can of course be reproduced, but not necessarily its instantaneous realizations.

**The discretised equations are deterministic and thus reproducible if the same initial conditions are used. The non-reproducibility of the instantaneous realizations is due to random perturbations that are set at the beginning of the simulation to "initialise" the turbulence. In our case, the turbulence is initialised with a spin-up simulation, that is used as the initial condition for both the main simulation and the reference simulation. These two simulations thus use the same initial conditions and should therefore be similar at each time step, except at the turbine's location.**

- Figure 8,9,10,12: "Your results do not seem to be that well converged". Additional discussion on the uncertainty in your data is required.

**This is indeed a flaw of our study, that we acknowledged only when post-processing the results. We added at the end of Sect 5.1:**

*"For all cases, the non-zero values of each term in the near wake (first column of every figure) are mostly distributed around the tip of the blades. For pure-terms (III) and (IV), they are spatially smoothly distributed at $x/D=5$ and $x/D=8$. For cross-terms (V) and (VI) and (VII), the non-zero values at these positions are chaotically distributed spatially and thus harder to interpret due to a lot of small-scale variations. A statistical averaging of every term over several simulations could provide data with better spatial coherence and the different terms would thus be easier to interpret. To do so, longer simulations with similar mean upstream conditions are needed."*

- Given that you want to use the results for model development, should you not determine the developments of the different terms further downstream the wind turbine. Now you only go to x/D=8.

**Ideally, the study should have been carried on up to x/D=12 or 15. There are two main reasons for this choice of restraining to x/D=8: first, a longer domain would require more computational resources for the LESs, which were already quite expensive. Moreover, the wake tracking is less and less reliable as the wake travels downstream. With the algorithm initially chosen, we knew that there would be a lot of errors at x/D=8 and did not consider computing the wake further. Meanwhile, we improved the tracking algorithms [1] and such a study would now be possible but would have required to re-launch the simulations, which we could not afford.**

**We thus added the following paragraph before in Sect 3.3:**

*"The size of the domain of interest ($D_4$ in the neutral and unstable case and $D_2$ in the stable case) is set to compute the wake up to $8$ diameters downstream the turbine. This choice has been made to keep reasonable simulation times for the LES and a high degree of confidence in the wake tracking algorithm. However, the wake is not dissipated at this position, and the present work could be completed with a study where the wake is computed further downstream, e.g. $x/D=15$."*

-- 375: the statement seems rather bold. Only 3 cases are presented, and the added

turbulence intensity for the three cases is not really the same. Looking at figure 12; A difference of 3 to 4 percentage points on 16 percent turbulence intensity.

**This statement has been removed. It was indeed a bit bold, and anyway out of the scope of the paper (it is more deeply studied in the aforementioned paper). We thus replaced the first lines:**

*"First, it must be noted that the magnitude of the normalised added turbulence in the vicinity of the turbine (at x/D = 1) is very similar in all cases (between 15 % and 18 %), despite different values of atmospheric stability, shear and hub height velocity. At this position, the added turbulence in the MFOR is almost axisymmetric. Since the thrust coefficient and tip speed ratio are similar for the three cases, it seems acceptable for future model calibrations to suppose that ΔkMF is solely a function of the turbine regime."*

**With:**

*"As shown in Jézéquel et al. (2022), the atmospheric stability mostly affects meandering and not the field in the MFOR: the magnitude of the normalised added turbulence in the vicinity of the turbine (at x/D = 1) is about 18 % in the neutral and unstable case, and the slightly lower value in the stable case (about 15 %) is attributed to smaller length scales upstream the turbine. As the wake travels downstream, the asymmetry increases, in particular for the neutral and stable cases, but the magnitudes of ΔT I are still similar among the different cases despite different values of atmospheric stability, shear and hub height velocity.*

- Line 391: How about figure 10? It is not quite clear what the statement "that it is reachable given the shapes." Just before you state "in the wake with the term (V) could be found yet." I tend to agree with the latter statement. I do not see any particular patterns

**This latter statement has been replaced with:**

*"If more data were available, one could perform an ensemble average and hopefully find a shape easier to interpret for this term."*

- Figure 12: How does the rotor added turbulence in the moving reference frame compare to what you would get in the fixed reference frame. What is your uncertainty in these results? There namely seems to be significant rotor added turbulence outside the wake.

**The same graph, as Fig. 12 for the FFOR is displayed below. One can see that the shapes of the field are much more different between the different cases than in the MFOR. About the rotor added turbulence outside the wake (for instance on the right of the neutral case), this appears in both frames of**

**reference, and is here attributed to a more energetic ABL eddy located at this position in the turbine simulation compared to the reference simulation. Once again, we think that an ensemble average of several datasets could improve these results, but we did not quantify the related uncertainty.**

[Figure]

* Typos

- Line 94: translation "opertor" --> "operator"; "time-dependant" --> "time-dependent"

- Velocity in the wake --> you are not actually showing velocities, but streamwise turbulence intensity.

**Thank you for the typos, they have been corrected.**

**RC2:**
General comments:

The paper presents an in-depth investigation into the specific mechanisms behind the mean velocity and turbulence following a wind turbine in a fixed frame of reference. The study breaks down the contributions and discusses the importance of each term within a stable, neutral, and unstable regime. By discussing the relative importance of each component's contribution to the mean velocity and the

turbulence, conclusions on what needs to be considered when modeling can be made and can be discussed specific to the stability regime.

The importance of the terms that contribute to the mean velocity and turbulence are found based on a large eddy simulation with an actuator line method. The study investigates the wake of a single wind turbine from near-field x/D=1 to far-field x/D=8 locations.  The referee believes the current work is of interest to those modeling turbine wakes and should be considered for publication after revisions. Comments are specified below.

Specific comments:

There are a points in the introduction when a citation should be included to validate the statement of the authors, for example, line 27 sentence beginning 'Most analytical models..'

**We added a reference to [2] for the velocity and [3] for the added turbulence. Actually, we could have cited any analytical model to illustrate this, except the DWM or the model of Braunbehrens and Segalini 2019 [4].**

The coordinate system should be explicitly stated, perhaps placed in figure 1, to orient the reader initially.

**This has been added to Fig. 1 and the sentence preceding Eq. 1 has been modified as:**

**"If a Cartesian coordinate system (x, y, z) is used for the streamwise, lateral and vertical coordinates (see Fig. 1), the instantaneous streamwise velocity can be changed from one frame to another according to the relation:"**

Why does the study only look at the wake until 8D downstream (Computational cost, based on current farm arrangements, etc.?)

**Please refer to our answer to the 4th comment of RC1 that was similar.**

Line 267 – The authors discuss errors in the unstable case only at x/D = 8 (at 6%) but leave out the error at x/D = 3, which is touched on as a location where the neutral boundary layer flow case shows reasonable overestimation at 2%. Comparisons should be discussed at this location as well as farther downstream at 8D because this effects turbine placement downstream.

**We replaced:**

**"The relative error is larger in the unstable case (reaching about 6 % at 8D downstream), and much lower in the stable case (less than 0.3 %)."**

**With:**

*"The relative error is larger in the unstable case, going from +5 % to -6 % between 1D and 8D downstream. This negative value shows an underestimation of the mean velocity by term (I) in the far wake. One can note that, at these positions, the tracking algorithm of the unstable case is less reliable, so it could be the source of the error. In such a case, approximating U_FF with (I) would be correct and the error would come from our methodology. In the stable case, the error is much lower: less than 0.3 %."*

Lines 225-229 – If velocity is stored at a rate of 1Hz, then are the first and second order statistics averaged over, for example for the stable case, only 60 snapshots? Uncertainty of the turbulence breakdowns and convergence is not discussed.

**The stable case is 10-minutes long, so it makes 600 snapshots. We acknowledge that this could be improved with more data (in particular for the unstable case). A higher sampling frequency could indeed improve the results but would have required to re-start the simulations, which was not affordable in this project. Nonetheless, we agree with the referee that more data would allow us to compute ensemble average and improve the statistical convergence of our results.**

**We added the following paragraph in the conclusion:**

*"The statistical convergence of the data could not be assessed but it is clear to the authors that the results of this work could be improved with better-sampled data. In particular, the present work suggests that 40-minutes averages are not sufficient to get converged data for wakes in unstable ABL. Increasing the simulation time would allow computing an ensemble average of the different terms, which will then be expected to be easier to interpret."*

Lines 225-229 – More information needs to be included on how the contributions of sub-grid scales are quantified and in turn negligible as the authors suggest.

**A 1.5-order closure is used in Meso-Nh to compute the subgrid fluxes: the subgrid TKE is a prognostic variable likely to u, v, w or $\theta$. We can thus compute a ratio between the subgrid TKE and the total TKE (i.e. subgrid + variances of u, v and w). Averaged on each level of the domain, it gives distributions like in the figure below for the different nested domains (here for the neutral case). Note that only domain D4 (orange) is of interest, as the other domains serve only to generate realistic boundary conditions for D4. In the figure (neutral case), except near the ground, the subgrid TKE is about 2-5% of the total TKE. In the unstable case, it is even less (because there is a lot of large-scale**

**turbulence) but in the stable case, results are not so good due to the absence of large-scale turbulence, and the subgrid TKE can reach more than 20% (and not 10% as firstly indicated) of the total turbulence.**

[Figure]

**Line 230 as thus been changed from:**

*"The latter is negligible in the unstable and neutral cases but can reach more than 10% in the stable case"*

**To:**

*"Since subgrid turbulence is a prognostic variable in the 1.5-order closure used in Meso-NH, one can compute the ratio between subgrid and total turbulence. It is between 1 % and 5% in the neutral and unstable case but can reach more than 20 % in the stable case. This highlights the difficulty of simulating strongly stratified ABL, but our results have been successfully compared to the SWiFT benchmark (Jézéquel et al., 2021), so they will be used nonetheless "*

Figure 7 shows the RSME of the maximum axial turbulence, is there an explanation for the max RSME value to be at x/D = 5 for the neutral case (is it also observed in other cases)? Also, no quantitative information is provided for the other two stability cases, only trends of the data and order of magnitude.

In the unstable case, a similar maximum error appears at 6D and then the error of (III)+(IV) decreases. To the authors' interpretation, this comes from an error compensation, combined with the fact that in the far wake $k_{x,LES}^{M}$ is low and thus the computed value is very sensitive to small changes. The authors hesitated to display the following figure instead of Fig. 7, where it can be easier to understand what happens as the different terms are added, but is a bit redundant with Fig. 6, takes more place and is more qualitative. On this graph we see that at x/D=8, the maximum TKE of (III)+(IV) in the restricted area ([-2D,2D],[-1D,1D]) is at the bottom of the domain due to edge effects. This decrease of error observed in Fig.7 is thus attributed to edge effects rather than actual better performance.

[Figure]

We have thus added the following precision line 289:

*"In the far wake (beyond $x/D=5$) the error decrease but this is a numerical artefact: due to edge effects, large TKE values are observed near the ground, and thus the maximum TKE is detected at this location instead of at the top tip."*

Technical corrections:

Line 94 – opertor => operator

Line 96 – ...U_MF allows to re-write Eq. 1... => ...U_MF allows one to re-write Eq. 1...

Line 134 – ...in Sect. 2 is applied... => ...in Sect. 2 are applied...

Line 321 - ...turbulence is going back its unperturbed value.. => ...turbulence is going back to its unperturbed value..

**Thank you for the typos, they have been corrected.**

**References:**

[1] Jézéquel, E.; Blondel, F. & Masson, V. Analysis of wake properties and meandering under different cases of atmospheric stability: a large eddy simulation study Journal of Physics: Conference Series, IOP Publishing, 2022, 2265, 022067

[2] Fuertes, F. C.; Markfort, C. & Porté-Agel, F. Wind Turbine Wake Characterization with Nacelle-Mounted Wind Lidars for Analytical Wake Model Validation Remote Sensing, 2018, 10, 668

[3] Ishihara, T. & Qian, G.-W. A new Gaussian-based analytical wake model for wind turbines considering ambient turbulence intensities and thrust coefficient effects Journal of Wind Engineering and Industrial Aerodynamics, 2018, 177, 275-292

[4] Braunbehrens, R. & Segalini, A. A statistical model for wake meandering behind wind turbines Journal of Wind Engineering and Industrial Aerodynamics, 2019, 193, 103954

---

## Author Response (AR2)

**Reviewer #1**

In the revision, the authors only partially addressed the comments in my referee report. One of the main comments "Although the statistical convergence of the data seems limited and uncertainty analysis is limited. These aspects should be improved." Of my previous report was essentially not addressed and this should still be done.

---> While the authors acknowledge that the statistical convergence could have been better, no further data on this is required. Instead, the authors state that no additional statistical data could be obtained, as the simulations could not be extended as these are computationally expensive [give the simulations under consideration I can see it is not easy to run these much longer]. However, the authors would not have to perform additional simulations. Instead, they can provide a more thorough statistical simulation analysis of the already available data (I do not agree with the statement that "the statistical conference of the data could not be assessed", as standard statistical methods could be employed on the dataset available as is). In essence, the reader should get some feel for what the uncertainty in the presented data is.

Thank you for your feedback. We thus propose an analysis of the statistical convergence to answer both questions:

- Is the sampling frequency (1Hz) high enough?
- Is the segment length (respectively 80, 40 and 10 minutes for the neutral, unstable and stable cases) long enough?

To answer these questions, we respectively:

- Computed the 95% uncertainty interval for each term of the TKE breakdown equation as well as the MFOR turbulence.
- Separated each segment into two equal sub-segments and computed the corresponding values of each term. If the two sub-segment give similar results than the full segment, it means that increasing further the segment length is not expected to have any significant effect.

If this does not fit your expectations, could you please provide an example of what you mean by "standard statistical methods", if possible, on similar work. We added at the end of section 3:

The error induced by the choice of the \$1\$ Hz sampling has been estimated with a 95 \% confidence interval and discussed in Appendix \ref{Appendix:A} for the MFOR turbulence and all the terms of Eq. \ref{eq:TKEFFOR2MFOR}. In short, the chosen sampling frequency of 1 \$Hz\$ is sufficient for the neutral and unstable case. In the stable case, due to the higher share of small-scale turbulence, the 95 \% interval is large, indicating that a higher sampling frequency would improve the accuracy of the LES.

Due to numerical limitations, the segment lengths were constrained to 80, 40 and 10 (see Table \ref{table:NumParam}). To estimate the statistical convergence, each of these segments is divided in two and the difference between the two sub-segments and the full-simulation is assessed in Appendix \ref{Appendix:A}. It appears that the stable and the neutral simulations would barely benefit from an extension of their duration whereas the unstable simulation seems to change significantly between the two sub-segments of 20 minutes. This is particularly true for terms (V)

**and (VI) of the turbulence breakdown equation (Eq. \ref{eq:TKEFFOR2MFOR}). The appendix can be found in the paper attached.**

\*\* In the conclusions a paragraph should be added that discusses the potential impact of the statistical convergence on the results.

The paragraph on statistical convergence has been replaced with-:

"The statistical convergence of the data has been assessed and showed that increasing the sampling frequency would most likely improve the reliability of the stable case but would have a low impact for the two other cases. On the other hand, increasing the simulation time would probably change the unstable results but have few effects for the other cases. The uncertainty is the highest on the cross-terms of the turbulence breakdown equation, but the pure-terms are subject to only small uncertainty. For a better interpretation of these terms, it may be important to perform ensemble simulations to get reliable fields."

\*\* Line 400, a new comment has been added

"As shown in Jézéquel et al. (2022), the atmospheric stability mostly affects meandering and not the field in the MFOR."

--> I am not sure what this comment is precisely based what this comment is based on, but I disagree. For example, in figure 12, we can see that the wake shape is very different in the stable case than in the neutral and unstable case. In this context, it would be helpful to the Fixed frame of Reference results corresponding to figure 12, so the reader can compare.

**Indeed, this statement is more understandable if one looks at the FFOR turbulence field:**

However, since we already added 3 figures for your remark on figure 5 and some appendices for the statistical convergence, we did not want to add too many plots. Especially since this part is not the

core of the work. We still modified the corresponding paragraph, and the reader can report to our conference paper for more information. Or even to the zenodo deposit if needed.

Despite strongly different values of  $\Delta(IV)$  among the different cases, Fig. 12 show that the atmospheric stability mostly affects meandering and not the field in the MFOR. Indeed, the magnitude of the normalised added turbulence in the vicinity of the turbine (at x/D = 1) is about 19 % in the neutral and unstable case, and the slightly lower value in the stable case (about 15 %) is attributed to smaller integral length scales upstream the turbine. These small discrepancies must be compared to the values achieved in the FFOR: around 22%, 27 % and 16% for the neutral, unstable and stable case (Jézéquel et al., 2022). The skewed shape of the stable case is attributed to the veer that appears in such ABL, but is negligible in neutral and unstable ABL. As the wake travels downstream, the asymmetry increases, in particular for the neutral and stable cases, but the magnitudes of  $\Delta T$  I are still similar among the different cases despite different values of atmospheric stability, shear and hub height velocity. The asymmetry is attributed to the ambient shear, which increases with atmospheric stability. Negative values of  $\Delta k_{MF}$  are observed in the near wake between the wake centre and the edge in the neutral and unstable cases and also in the bottom of the far wake in the neutral case. This indicates a transfer of energy from such regions to the high turbulence region, i.e. the edge and the top of the wake. Overall, this figure shows that the different values of  $\Delta(IV)$  among the cases mainly come from the meandering operation and only slightly from the MFOR turbulence itself.

\*\* Line 458: "to correct this error and drastically reduce the overall RMSE." --> Yes adding the cross terms improves the results. However, figure 6 shows that in the direct wake region also using just terms III and IV approximates the turbulent profile in the wake well. The wording "Drastically reduce" is too strong given that you state that the results for the cross terms are not statistically converged.

**The word drastically has been removed.**

\*\* Figure 1 does not provide a sketch of the moving frame and mixed frame definitions suggested around line 40.

The picture has been changed to clarify the two frames. Moreover, the colors have been changed to improve the readability for colorblind persons.

---

## Author Response (AR3)

Thank you for accepting the manuscript. My answer to you last comments are written in blue:

Line 51 - Time is only taken into account in the left hand side of the MF equation, please revise.

To be more clear, the equation and the sentence around it have been replaced with:

*If the unsteady FFOR velocity field is required, Eq. 1 is used with a steady, axisymmetric form in the MFOR, i.e. UF F (x, y, z, t) =UM F (x, y − yc(x, t), z − zc(x, t))*

Line 86 - The sentence should read 'LES datasets' instead of 'LESs datasets'

This has been modified

Line 254 - The term segment length is introduced by not clearly defined, please clarify the term for the reader when it is introduced. Especially because the article refers to table 1, which gives the quantity in seconds, but uses minutes in the definition without acknowledging its units.

We replaced "segment length" with "duration of the simulation" and added the unit *min* that was missing in the text

Figure 8 caption - Replace the line ' different lines' with ' different rows' to better orient the reader to the figure.

This has been modified

Line 528 - 'excepted for term' should read 'except for term'.

This has been modified